# Applying FP_ILM to the retrieval of geometry-dependent effective Lambertian equivalent reflectivity (GE_LER) daily maps from UVN satellite measurements

Diego G. Loyola[1], Jian Xu[1], Klaus-Peter Heue[1], Walter Zimmer[1]

[1]German Aerospace Centre (DLR), Remote Sensing Technology Institute, Oberpfaffenhofen, 82234 Wessling, Germany

*Correspondence to*: Diego Loyola (Diego.Loyola@dlr.de)

**Abstract.** The retrieval of trace gas, cloud and aerosol measurements from ultraviolet, visible and near-infrared (UVN) sensors requires precise information on surface properties that are traditionally obtained from Lambertian equivalent

reflectivity (LER) climatologies. The main drawbacks of using LER climatologies for new satellite missions are (a) climatologies are typically based on previous missions with significantly lower spatial resolutions, (b) they usually do not account fully for satellite viewing geometry dependencies characterized by bidirectional reflectance distribution function (BRDF) effects, and (c) climatologies may differ considerably from the actual surface conditions especially with snow/ice scenarios.

In this paper we present a novel algorithm for the retrieval of geometry-dependent effective Lambertian equivalent reflectivity (GE_LER) from UVN sensors; the algorithm is based on the full-physics inverse learning machine (FP_ILM) retrieval. Radiances are simulated using a radiative transfer model that takes into account the satellite viewing geometry and the inverse problem is solved using machine learning techniques to obtain the GE_LER from satellite measurements.

The GE_LER retrieval is optimized not only for trace gas retrievals employing the DOAS algorithm, but also for the large

amount of data from existing and future atmospheric Sentinel satellite missions. The GE_LER can either be deployed directly for the computation of AMFs using the effective scene approximation or it can be used to create a global gapless geometry-dependent LER (G3_LER) daily map from the GE_LER under clear-sky conditions for the computation of AMFs using the independent pixel approximation.

The GE_LER algorithm is applied to measurements of TROPOMI launched in October 2017 on board the EU/ESA Sentinel-

5 Precursor (S5P) mission. The TROPOMI GE_LER/G3_LER results are compared with climatological OMI and GOME-2 LER datasets and the advantages of using GE_LER/G3_LER are demonstrated for the retrieval of total ozone from TROPOMI.

## 1. Introduction

Lack of knowledge of the magnitude of surface reflectance and the neglect of surface anisotropic effects are the two major error sources for the retrieval of trace gas, cloud and aerosol information from ultraviolet, visible and near-infrared (UVN) satellites measurements (Vasilkov et al., 2018; Lorente et al., 2018; Lin et al., 2014; Seidel et al., 2012; Zhou et al., 2010). Surface reflectance has a stronger influence on the retrievals of boundary layer trace gases and aerosols than is the case for mid- and upper-tropospheric trace gas and cloud retrievals. For example errors of 0.02 in the surface reflectivity may induce errors of 10%−20% in retrieved $SO_2$ column amount (Lee et al., 2009) and seasonal snow cover can change the retrieved $NO_2$ column by 20%−50% (O'Byrne et al., 2010) and the retrieved $O_3$ column by 5%−35% (Lerot et al., 2014).

The Lambertian Equivalent Reflectivity (LER) concept was first introduced for the BUV (Backscatter Ultra-Violet) total ozone retrievals (Mateer et al., 1971) and it was extended to retrievals of tropospheric ozone, $NO_2$, $SO_2$ and other pollutants under partly cloudy conditions using the independent pixel approximation (Ahmad et al., 2004). Traditionally, surface properties are obtained from LER climatologies and in the case of new missions such as TROPOMI launched in October 2017 on board the EU/ESA Sentinel-5 Precursor (S5P) mission, climatologies used at mission start are based on LER data from previous missions such as TOMS (Herman and Celarier, 1997), GOME (Koelemeijer et al., 2003), OMI (Kleipool et al., 2008), SCIAMACHY (Tilstra et al., 2017), and GOME-2 (Pflug et al., 2008).

The unprecedented spatial resolution of TROPOMI (3.5 x 5.5 km² currently and 3.5x7 km$^2$ for data before August 6[th] 2019) has clearly shown the disadvantages of using LER climatologies based on previous missions with significantly lower spatial resolution. Indeed, initial studies of the TROPOMI trace gas retrieved products based on such LER climatologies have exhibited flawed patterns related to the coarser resolution of the OMI LER climatology. A LER climatology based on TROPOMI measurements could solve this particular problem, but creating a new TROPOMI LER climatology will probably require at least two years of data. Furthermore, there are two common problems with typical LER climatologies: (a) the actual surface conditions of a satellite measurement may differ considerably from climatological values, as for example for snow/ice scenarios, and (b) the effect of surface reflectance anisotropy is usually not properly covered by the climatology.

Retrieval of Lambertian effective scene albedo has been used in total ozone algorithms from nadir and limb satellite sensors, see for example Bhartia et al., 1996 and McPeters, et al., 2015. The WFDOAS (Coldewey-Egbers et al., 2005) algorithm retrieves the effective LER at 377 nm, while the GODFIT (Lerot et al., 2010) and SAGE III (Raul and Taha, 2007) approaches both retrieve simultaneously the effective LER and other parameters along with total ozone.

Another approach used for $NO_2$ and cloud retrievals involved the computation of LER from bidirectional reflectance distribution function (BRDF) data obtained from other satellite sensors with higher spatial resolution. In a recent work (Vasilkov et al., 2017), the BRDF data from MODIS is first resampled to the lower resolution of the OMI instrument, and then a geometry-dependent LER is computed using radiative transfer model simulations. Unfortunately MODIS BRDF data is available only from visible (VIS) wavelengths and rescaling the VIS BRDF (or LER) to UV is not straightforward.

Furthermore, the radiative transfer (RT) model assumptions needed for computing MODIS BRDFs may not be fully compatible with RT model assumptions required for UV-based trace gas retrievals.

In this paper we present a novel algorithm to be used not only for the retrieval of geometry-dependent effective Lambertian equivalent reflectivity (GE_LER) from UVN measurements but also for the creation of global gapless geometry-dependent

LER (G3_LER) daily map based on GE_LER data obtained for clear-sky conditions. The retrieved GE_LER and G3_LER should represent the current surface conditions, while mitigating the problems of using LER climatologies, and accounting for surface anisotropy effects in cloud, aerosol and trace gas retrievals, in a similar manner as does the effective LER (Coldewey-Egbers et al., 2005) and the geometry-dependent LER (Qin et al., 2019). But in contrast to these approaches, the GE_LER retrieval is performed in precisely the same fitting windows used for the trace gas, cloud and aerosol retrievals

themselves; furthermore our algorithm does not require external data sources such as BRDFs (land surfaces) or Chlorophyll and wind parameters (water surfaces).

First we describe in Section 2 the full-physics inverse learning machine (FP_ILM) technique used for the retrieval of GE_LER from UVN measurements, and we demonstrate how it is optimized for DOAS trace gas retrievals. Section 3 discusses the creation of global gapless geometry-dependent LER (G3_LER) daily maps using the retrieved GE_LER for

clear-sky conditions. In section 4 we apply the GE_LER algorithms to S5P measurements, first comparing TROPOMI G3_LER results with climatological OMI and GOME-2 LER data, and secondly we demonstrate the advantages of using GE_LER/G3_LER for the retrieval of total ozone from TROPOMI. In Section 5 we discuss future work.

## 2. The FP_ILM algorithm for the GE_LER retrieval

Trace gas, cloud and aerosol retrievals from UVN measurements rely on complex radiative transfer model (RTM)

simulations. RTM calculations are computationally expensive and therefore not well suited for processing massive data from the new generation of atmospheric-composition Sentinel missions. A classical approach for speeding up RTM performance is to use look-up tables (LUTs). The main drawbacks of LUTs with high dimensionality (common in atmospheric composition retrievals) are that the memory requirements increase exponentially with the number of input dimensions, the interpolation/extrapolation in this multi-dimensional space are computationally expensive, and interpolation/extrapolation

errors can be significant. To avoid these LUT issues, the DLR team has developed during the last two decades machine learning techniques for the optimal generation of RTM samples (Loyola et al., 2016) and the accurate parameterization of RTM simulations using artificial neural networks (NN). These algorithms are being used for the operational processing of GOME-2 (Loyola et al., 2010) and now TROPOMI (Loyola et al., 2018) data.

Machine learning can be used not only for forward problems (such as the parameterization of RTM simulations), but also for

solving inverse problems, see for example (Loyola et al., 2016). Recently we have developed an approach called the "full-physics inverse learning machine" (FP_ILM) technique; this has been applied successfully for retrieving ozone profile

shapes from GOME-2 (Xu et al., 2017) and retrieving SO$_2$ layer height from GOME-2 (Efremenko et al., 2017) and TROPOMI (Hedelt et al., 2019).

**Figure 1** presents a flow diagram of the different steps of the FP_ILM algorithm and the following subsections describe in more detail how FP_ILM is tailored for the retrieval of GE_LER.

## 2.1. Forward Model

The forward model segment has two components: first a radiative transfer model (RTM) that computes spectral intensity as a function of the solar and viewing geometry, atmospheric components and Lambertian surface properties; and second a sensor model that transforms the RTM intensities to simulated spectra using sensor information such as the instrument spectral response function and signal to noise ratio.

The forward model $F$ will simulate spectral radiances $R_{sim}$ for a given wavelength $\lambda$ according to

$$R_{sim}(\lambda) \pm \varepsilon_R = F(\lambda, \Theta, \Omega, A_e, Z_e) \tag{1}$$

where $\varepsilon_R$ denotes the expected instrument error, $\Theta$ is the light path geometry (solar and satellite zenith and azimuth angles), $\Omega$ are the atmospheric composition components, and the surface properties denoted by $A_e$ for the geometry-dependent effective Lambertian equivalent reflectivity (GE_LER) and $Z_e$ for the effective surface height.

## 2.2. Smart Sampling

Traditionally, training data are created at uniformly distributed fixed intervals for each input variable; as a consequence, the training samples are grouped around the node points and poor coverage of the multidimensional input space is the result. Deterministic sampling methods provide a more uniform distribution of the training data covering the entire space of each input variable.

A key element of FP_ILM is the creation of a training data set that covers extensively the multidimensional space of the forward problem and at the same time minimizes the computational expensive calls to the radiative transfer model. We use the smart sampling techniques (Loyola et al., 2016) for creating a dataset of samples $\{\Theta, \Omega, A_e, Z_e\}$ that fully represent the expected viewing and geophysical conditions of the problem at hand. For this work we select a Halton sequence that uses prime numbers for creating sample points in each input dimension and a RTM that computes the corresponding simulated radiances.

As indicated in **Figure 1**, the smart sampling and forward module calls are iterated in a loop until the multi-dimensional integral of the output samples dataset $\{R_{sim}(\lambda) \pm \varepsilon_R\}$ converges. This technique allows us to determine the minimum number of samples needed to properly cover the output space; see (Loyola et al., 2016) for more details.

## 2.3. Feature Extraction

Retrieval of trace gas, cloud and aerosol concentrations from UVN sensors requires spectrometers with sufficiently detailed spectral resolution to resolve absorbing features in the electromagnetic spectrum. The fitting window used for retrieval of a trace gas usually requires hyperspectral radiances for a high-dimensional space (tens to hundreds of wavelengths). Machine learning techniques perform best with low-dimensional datasets.

Feature extraction is a mapping function that transforms a dataset from a high- to a low-dimensional space by the removal of redundant information and noise. In previous FP_ILM applications (Loyola et al., 2006; Xu et al., 2017) we used principal component analysis for the feature extraction. However for the GE_LER retrieval we take advantage of the DOAS fitting model

$$R_{sim}(\lambda) = -\sum_g N_{s,g}(\Theta) \cdot \sigma_g(\lambda) - p(\lambda) \tag{2}$$

where $N_{s,g}(\Theta)$ is the effective slant column density of gas $g$ for light path geometry $\Theta$, $\sigma_g(\lambda)$ the associated trace gas absorption cross-section at wavelength $\lambda$, and $p(\lambda)$ an external closure polynomial.

The feature extraction step comprises an application of the DOAS fit to the simulated radiances. Combining (1) and (2) for a given fitting window $\Lambda$ we obtain the following approximation with simulated datasets that represent the forward problem

$$\{N_{s,g}(\Theta), P(\Lambda)\} \cong \{F(\Theta, A_e(\Lambda), Z_e)\} \tag{3}$$

where $A_e(\Lambda)$ is the wavelength independent GE_LER for the particular DOAS fitting window.

### 2.4. Machine Learning

Machine learning approximates a function that is represented by input/output datasets by means of linear or non-linear regression algorithms. In this paper we use artificial neural networks (NN) to learn the non-linear inverse problem by reorganizing the datasets from (3) to represent the inverse problem:

$$\{A_e(\Lambda)\} \cong \{F_{NN}^{-1}(p(\Lambda), N_{s,g}, \Theta, Z_e)\} \tag{4}$$

In other words, a neural network will solve the inverse problem and retrieve the GE_LER as function of the DOAS closure polynomial, the DOAS fitted effective slant column density, the viewing geometry and the effective surface height. The inverse operator itself is the collection of the weights and biases of the neural network approximating $F_{NN}^{-1}$.

### 2.5. GE_LER Retrieval

Obtaining the inverse operator is very time consuming mainly due to the relatively large amount of RTM simulations needed to properly characterize the forward problem. Finding a neural network (NN) topology that learns the inverse function with minimum error is also computationally intensive. However, all these steps are carried out offline and need to be done only once for a given sensor and trace gas fitting window.

Figure 2 shows the flow diagram for applying the FP_ILM to satellite measurements. There is no additional computation needed for the feature extraction part, as we are using results from the DOAS fitting; also, application of the NN to retrieve GE_LER is very fast as it only involves simple matrix multiplications.

The exceptionally fast retrieval using the FP_ILM is a crucial advantage for the operational near-real-time processing of the Big Data from current and future atmospheric composition Sentinel missions.

## 3.   Global Gapless Geometry-dependent (G3) LER Daily Map

Conversion of DOAS effective slant column amounts to geometry-independent total column requires the calculation of air mass factors (AMF) calculated using either the effective scene approximation (Mateer et al., 1971; Coldewey-Egbers et al., 2005) or the independent pixel approximation (e.g. Loyola et al., 2011). The retrieved GE_LER can be used directly for AMF computation based on the effective scene approximation; clear-sky LER is needed for AMFs calculated with the independent pixel approximation.

A global gapless geometry-dependent LER (G3_LER) daily map can be easily created from GE_LER values retrieved under clear-sky conditions. In the case of S5P, a clear-sky situation is established not only with the operational cloud properties retrieved from TROPOMI (Loyola et al., 2018) but also with the VIIRS/SNPP (Visible Infrared Imaging Radiometer Suite sensor, on board the Suomi National Polar-orbiting Partnership satellite) cloud mask regridded to the TROPOMI spatial resolution (R. Siddans, 2016). Note that S5P and SNPP fly in loose formation, with the S5P orbit trailing 3 to 5 minutes behind SNPP.

The G3_LER map for a given day is created by merging the clear-sky GE_LER data from the same day with the G3_LER map based on the GE_LER data from previous days, see Figure 3. The spatial resolution of the G3_LER maps for TROPOMI is 0.1° latitude and 0.1° longitude, and global maps can in general be derived by combining data from a single month. Two to three months of data are needed only for regions with persistent cloud cover such as the Intertropical Convergence Zone (ITCZ).

It is important to note that the GE_LER determination incorporates bidirectional reflectance distribution function (BRDF) effects, since it is based on radiative transfer model simulations using the actual viewing geometry. When combining GE_LER data their BRDF dependencies $\rho(\Lambda, \theta, \psi)$ as function of the wavelength in the fitting window $\Lambda$, the viewing zenith angle $\theta$, and the surface type $\psi$ must be considered. In contrast, solar zenith angle dependencies can be ignored when combining GE_LER data from sun-synchronous satellites over the same location, because the angle of sunlight at the Earth's surface is consistently maintained. Likewise relative azimuth angle dependencies are negligible in the UV. The $\rho(\Lambda, \theta, \psi)$ dependencies can be obtained separately for different fitting windows $\Lambda$ (in the UV, VIS and NIR spectral region), for different surface types $\psi$ (e.g. land, water, snow/ice) and various time periods (e.g. monthly); any dependency on viewing

zenith angle can be characterized by fitting a polynomial (or exponential) function over clear-sky LERs sorted as function of $\theta$.

The G3_LER daily map comprises the normalized LER, i.e. the GE_LER retrieved under clear-sky conditions divided by the fitted BRDF dependency, as well as the multiplicative factors $\rho(\theta)$ needed to compute the geometry-dependent LER as a function of the actual satellite viewing zenith angle $\theta$.

It is necessary to aggregate normalized LER retrievals over several days (between one to four weeks depending on cloudiness) in order to obtain a global gapless map. In contrast to LER climatologies, the G3_LER map represents actual surface properties as it is updated on a daily basis. The only exceptions are cases of sudden snow fall combined with significant cloudiness.

## 4. GE_LER and G3_LER from TROPOMI/S5P 325-335 nm

In this section, we apply the GE_LER and G3_LER algorithms described in the previous sections to measurements of TROPOMI/S5P in the total ozone wavelength region. The S5P operational near-real-time total ozone products (Loyola et al., 2019) are based on the DOAS algorithm with fitting window 325-335 nm. First we discuss aspects of the training process.

### 4.1. GE_LER Training

The training dataset is based on spectra simulated by the Vector LInearized Discrete Ordinate Radiative Transfer (VLIDORT) model (Spurr, 2016). The RTM inputs are ozone concentration profiles, Lambertian surface albedo, surface height and the solar and viewing angles. The smart-sampling technique (Loyola et al., 2016) was used to create more than $2 \times 10^5$ synthetic UV spectra for the ozone profile, viewing geometry and surface parameters in the ranges listed in Table 1. We use the Bodeker et al., (2013) ozone profile climatology for representing the stratospheric ozone in conjunction with the McPeters/Labow (Labow et al., 2015) climatology for lower tropospheric ozone.

Synthetic TROPOMI/S5P measurements are created by convolving these RTM radiances with the instrument slit function and adding a Gaussian instrument noise with a signal-to-noise ratio of 300 representative of TROPOMI band 3, see Kleipool et al., 2018.

The DOAS fitting is applied to these simulated S5P radiances using the same DOAS settings as in the operational S5P retrieval including a cubic external-closure polynomial resulting in a dataset of ozone slant columns and associated polynomial coefficients.

Figure 4 shows the optical densities of the DOAS polynomial ($p(\lambda)$) in Equation (2) for three scenarios. In panel (a) these are given as functions of four typical values of surface albedo of 0.05, 0.3, 0.6, and 0.9 which correspond to water, land, melted snow/ice-cover and fresh snow/ice-covered regions. The largest absolute value of the optical density corresponds to the highest surface albedo; optical densities for the four albedos do not differ significantly at lower wavelengths, while the

differences are more significant at the longer wavelengths. In panel (b) optical densities of the DOAS polynomial are shown with respect to three total ozone columns of 150 DU, 300 DU, and 500 DU; the absolute value of the optical density increases when the total ozone column increases. Finally in panel (c) densities are plotted for three viewing zenith angles of 50°, 30°, 10°; the absolute value of the optical density increases with decreasing viewing zenith angle. For all cases, optical density increases with wavelength.

The simulation results from (3) are reorganized by grouping the DOAS polynomial coefficients, ozone slant column, the viewing geometries, and surface heights as inputs to the neural network. A feedforward neural network (the neurons are grouped in layers) is trained to learn the inverse function (retrieval of surface albedo) using 70% of the simulations for training, 15% for testing and 15% for validation. Different NN topologies were tested using one, two, and three hidden layers; the best results are obtained using a NN with a topology of 9-20-8-2-1, which is 9 neurons in the input layer, three hidden layers with the given number of neurons, and one neuron on the output layer.

In Figure 5, we depict the GE_LER retrieval errors as function of the different input parameters calculated using the validation dataset (i.e. part of the dataset not used for NN training); the x-axes are divided into bins and the mean and standard deviation (red bars) are calculated for each bin. Differences between the *true* and retrieved GE_LER are very small with a mean and standard deviation of only $0.0016 \pm 0.0018$. These results demonstrate that the NN represents the inverse function in a very accurate manner.

The Ring effect (filling in of Fraunhofer and telluric spectral signatures through inelastic rotational-Raman scattering by air molecules) is a significant spectral interference in DOAS total ozone fitting in the 325-335 nm window. We tested its impact for the GE_LER training by adding filling-in corrections obtained with the LIDORT-RRS model (Spurr et al., 2008) to the VLIDORT simulations. We found that the Ring-effect impact on GE_LER retrieval in the ozone fitting window is not significant. Indeed, the mean difference in GE_LER retrievals with and without the inclusion of LIDORT-RSS corrections is in the range of 5e-5 for SZA<75° and 3.5e-4 for larger SZA.

The BRDF effects on the ozone fitting window are well modelled using the GE_LER approximation, the difference in the total ozone retrieved using VLIDORT with and without the BRDF supplement is in the order of 0.5 DU or 0.2%.

## 4.2. GE_LER Retrieval

The neural network trained with the inverse function is applied to TROPOMI/S5P measurements. The inputs are the DOAS fitted polynomial coefficients and ozone slant column, the solar and viewing zenith angles, the relative azimuth angle, and the effective surface height $Z_e$ computed in the independent-pixel approximation as

$$Z_e = (1 - f_c)Z_s + f_c\, Z_c \tag{5}$$

where $f_c$ is the cloud fraction, $Z_s$ the surface height, and $Z_c$ the cloud height. The S5P cloud properties are obtained from the operational TROPOMI cloud products using the OCRA and ROCINN algorithms (Lutz et al., 2016; Loyola et al., 2018).

It is known that version 1 of the TROPOMI Level 1 product has small deficiencies in the UV band (Rozemeijer and Kleipool, 2019); therefore a "soft" correction based on comparisons with OMPS radiances is applied to the S5P radiances. It is expected that these issues will be solved for version 2 of the TROPOMI Level 1 product, obviating the need for this soft correction.

TROPOMI/S5P GE_LER results for the total ozone fitting window (325-335 nm) for April 10th, 2018 are shown in Figure 6. As expected the GE_LER field shows the same patterns as the cloud field for that day. For clear-sky conditions ($f_c \leq 0.05$) the GE_LER represents the hemispherical surface albedo, while for cloudy scenarios ($f_c \geq 0.95$) GE_LER represents the cloud albedo. Figure 7 shows histograms of the differences between the TROPOMI clear-sky GE_LER and the OMI LER climatology (Kleipool et al., 2008) and also the differences between the cloudy TROPOMI GE_LER and the cloud albedo from the operational cloud product retrieved with ROCINN_CRB (Loyola et al., 2018). The second mode around 0.5 in the histogram indicates snow- or ice-cover scenarios in TROPOMI data that are poorly represented with the OMI LER climatology. The comparison between S5P GE_LER and the GOME-2 and OMI climatologies is discussed in more detail in Section 4.4.

Mean differences for the clear-sky and cloudy cases as a function of the surface type are summarized in Table 2 and Figure 7, with the relatively larger offsets and spreads mainly due to the difference in spectral regions between GE_LER retrieved in the UV (325–335 nm) and the cloud properties retrieved with ROCINN_CRB from the $O_2$A-band in the NIR (758–771 nm).

### 4.3. G3_LER Daily Map

The TROPOMI G3_LER map for a given day is created by regridding (at resolution 0.1° x 0.1°) the clear-sky LER data from the same day with the G3_LER map based on LER data from previous days. The LERs are obtained from the S5P GE_LER retrievals under clear-sky conditions. In this version of the TROPOMI G3_LER map we use the S5P OCRA and the VIIRS/SNPP (flying in constellation with S5P) cloud fractions $f_c$ for identifying clear-sky measurements ($f_c \leq 0.05$ is the criterion here). In the future we plan to additionally use the S5P absorbing aerosol index product for an even more stringent cloud/aerosol screening.

Ground pixels affected by sun glint and solar eclipse are removed according to corresponding flags available in the S5P total ozone product (Pedergnana et al., 2018). The remaining LERs from a given day replace the corresponding grid points of the G3_LER map from the previous day. Time information (orbit number) of the LER used in each grid cell is included in the G3_LER maps.

The BRDF dependencies $\rho(\theta)$ are calculated by fitting a polynomial to the TROPOMI LER data normalized to the central detector pixel (nadir viewing) and averaged as function of the viewing zenith angle. Three different surface types are considered: land, water and snow/ice. Figure 8 shows the BRDF dependencies calculated with normalized TROPOMI/S5P

data from January, April, July and October 2018. The surface classification is based on the land/water mask and the snow/ice flags from the S5P total ozone product (Pedergnana et al., 2018). Note that these surface types are appropriate to BRDF effects in the UV ozone fitting window; other trace gases retrievals (such as $NO_2$ in the visible spectrum) will require different land cover types (e.g. water, snow/ice, urban, paddy, crop, deciduous forest, evergreen forest) to properly model
BRDF effects; see Noguchi et al., 2014.

Figure 9 shows the TROPOMI/S5P G3_LER daily map for April 30[th], 2018, plus a comparison with the OMI LER climatology for the month of April. The OMI LER climatology is based on 3 years of data (2004 to 2007) whereas the TROPOMI G3_LER is based on a few weeks of data. The main advantages of the TROPOMI G3_LER daily map compared to climatology are first that it better represents current surface conditions such as snow/ice contamination; second that it
accounts for BRDF effects; and third that it has improved spatial resolution (0.1°).

## 4.4.  G3_LER comparison with OMI and GOME-2 LER

In this section we compare TROPOMI G3_LER with climatology LER from OMI (Kleipool et al., 2008) and GOME-2 (Tilstra et al., 2017). Since TROPOMI G3_LER is retrieved with fitting window 325 to 335 nm, we chose 335 nm LER values from the two climatologies. For GOME_2 there is no shorter wavelength available in the published dataset, and for
the OMI climatology use of the 328 nm is not recommended (Kleipool et al., 2010). In the following the instrument names will act as synonym for the respective albedo data sets. The three albedo datasets have different time and horizontal resolutions: OMI covers four years with grid resolution 0.5°, GOME: covers six years with grid resolution 0.25°, and TROPOMI covers only one year with a grid of 0.1°.

The histograms in Figure 10 show the differences between the TROPOMI, OMI and GOME-2 albedo maps for three
different surface types land water and snow/ice. A grid cell is assumed to contain snow/ice if the albedo of all three instruments is above 0.7, and the latitude is outside the ±60° range. For the snow- and ice-free observations over land and sea, the latitude range was restricted to ±40°. In general the three data sets agree quite well. Over land and water the mean differences are lower than 0.03 and the distributions are small (standard deviation around 0.04). The histograms with S5P over land have tails towards higher values of up to 0.1 indicating that for some areas S5P data overestimate the albedo.
According to the corresponding world maps (Figure 11) this occurs mainly over rain forests in Brazil, central Africa or Indonesia, where the TROPOMI data might be affected by residual cloud contamination. Note that for TROPOMI we have only one year of data compared to the multi-years for OMI and GOME-2

Over snow and ice larger deviations are found between OMI and GOME-2 LERs and between TROPOMI G3_LER and the climatological OMI/GOME-2 LERs. We conclude that the historical climatologies from OMI and GOME-2 do not properly
represent actual snow/ice conditions observed in 2018/2019.

## 4.5.  Usage of TROPOMI/S5P G3_LER for the Total Ozone retrieval

The near-real-time S5P total ozone product is based on an iterative DOAS/AMF algorithm (Loyola et al., 2019) and the current operational version (1.1.7) uses the OMI LER climatology (Kleipool et al., 2008). The median bias between near-real-time total ozone from S5P and reference data from Brewer, Dobson, and SAOZ sites is of the order of +1% (Verhoelst et al., 2018; Garane et al., 2019).

S5P near-real-time ozone agrees well with the Copernicus Atmosphere Monitoring Service (CAMS) analysis with the exception of some anomalies at high latitudes (Inness et al., 2019). Those anomalies are associated with the coarse resolution of the OMI LER climatology and most importantly, with differences between climatological LER values and the actual surface conditions (mainly snow/ice).

When we replace the OMI LER climatology with the TROPOMI G3_LER daily maps, the resulting total ozone field is
significantly smoother and has significantly fewer outliers. Figure 12 shows the TROPOMI/S5P surface albedo and total ozone retrievals from April 1$^{st}$, 2018 around the Bering Strait separating Russia and Alaska. The TROPOMI G3_LER daily map agrees very well with the surface types apparent in the corresponding VIIRS/SNPP images (S5P flies only 3-5 minutes behind SNPP) including the water surface along the coastline of the Chukchi Sea in Russia, the Sarichef Island to the north of Alaska and the Seward Peninsula in south Alaska. These coastal water surfaces as well as the open water of the Bering
Sea are not well represented in the OMI LER climatology, which indicates snow/ice cover for these sea areas. Similarly, the OMI LER climatology (erroneously) shows no snow/ice cover in the Yukon–Koyukuk Census Area in Alaska. The coarse spatial resolution of the OMI LER climatology is clearly visible in the total ozone field, and in addition, incorrect snow/ice assignments in the OMI LER climatology induce large errors on the retrieved total ozone with differences between −10% and +15%.

Moreover, agreement of the S5P total ozone with the CAMS assimilation at high latitudes is significantly better than that for the LER climatologies, as seen in Figure 13. Mean differences between total ozone from S5P and CAMS for the complete month of April 2018 are summarized in Table 3. The agreement with CAMS improves considerably at all latitudes: differences in the total ozone for the region [80°S-60°S] are reduced from −2.61 ± 2.22% using OMI LER to 0.74 ± 2.43% using TROPOMI G3_LER, for [60°S-50°N] the difference remains at the same level with a small increase from
0.23 ± 1.14% to −0.38 ± 1.13%, in the region [50°N-70°N] is reduced from 1.24 ± 2.45% to −0.79 ± 1.98% and finally for [70°N -90°N] the difference is−1.001 ± 2.58% compared to −1.35 ± 2.5%.

## 5. Conclusions

We have developed a novel algorithm for the accurate and fast retrieval of geometry-dependent effective Lambertian equivalent reflectivity (GE_LER) from UVN sensors based on the full-physics inverse learning machine (FP_ILM)
technique. The main inputs to the GE_LER retrieval are the DOAS fitting polynomial coefficients and the fitted trace gas slant column amounts, as well as the satellite viewing geometry. The inversion problem is solved using neuronal networks

trained with radiative transfer model simulations based on the same kind of RTM and settings used for the AMF calculations.

A global gapless geometry-dependent LER (G3_LER) daily map can be created from the GE_LER retrievals under clear-sky conditions. The G3_LER daily maps better characterize current surface; in particular they minimize errors induced by the LER climatologies through inaccurate representation of snow/ice scenarios. Both GE_LER and G3_LER account for satellite viewing dependencies which are characteristic of BRDF effects.

GE_LER is retrieved from each single ground pixel using the same spectrum and DOAS/AMF settings as those employed for trace gas retrievals, and GE_LER is therefore fully consistent with the trace gas retrieval itself. This is in contrast to LER products based on data from other satellites or LER data derived from the same satellite but using different fitting-window or RTM settings. G3_LER maps are updated on a daily basis using the clear-sky GE_LER for that day, and they are evidently superior to LER climatologies that fail to represent actual surface conditions.

We have applied the FP_ILM algorithm to retrieve GE_LER from TROPOMI for the 325-335 nm fitting window and thereby generate daily G3_LER maps that are used to retrieve the S5P total ozone. S5P total ozone retrievals based on the TROPOMI G3_LER daily maps are clearly superior to those based on OMI_LER climatology. The ozone fields are not only smoother, but also the differences compared to the total ozone from CAMS in April 2018 is reduced from $-2.53 \pm 2.46\%$ to $0.78 \pm 3.49\%$ in the latitudinal region [80°S-60°S]. Errors in the S5P total ozone between $-10\%$ and $+15\%$ induced by snow/ice miss-representations in the OMI_LER climatology are removed with the FP_ILM GE_LER/G3_LER TROPOMI products.

GE_LER can be applied to any trace gas, cloud and aerosol product retrieved in the UVN and is fully compatible with the DOAS/AMF settings used for the trace gas retrievals. GE_LER and G3_LER can be used as inputs for computing AMFs, either with the effective scene assumption or the independent pixel approximation. In this paper we demonstrated their effectiveness for improving the quality of TROPOMI total ozone; in the near future we plan to extend GE_LER/G3_LER to fitting windows for the S5P operational UVN cloud product (Loyola et al., 2018), the UV/VIS trace gases $NO_2$ (van Geffen et al., 2018), $SO_2$ (Theys et al., 2017), HCHO (De Smedt et al., 2018) as well as to fitting windows for S5P research products such as $H_2O$, BrO, OClO, CHOCHO and aerosol optical depth.

The GE_LER retrieval is accurate and very fast and is therefore well suited for the (near-real-time) processing of massive data from the atmospheric Sentinel satellite missions. We plan to apply the FP_ILM GE_LER/G3_LER retrieval to the future Copernicus Sentinel-5 mission that (like Sentinle-5P) tracks along a sun-synchronous polar orbit. Furthermore, we plan to assess the suitability of GE_LER to capture the diurnal LER dependencies on the sun-satellite geometry of the future UVN geostationary missions Sentinel-4, TEMPO and GMES.

**Acknowledgements**

Special thanks to Robert Spurr for long-standing LIDORT support and editorial help. We thank the three reviewers for their insightful comments. This paper contains modified Copernicus Sentinel data processed by DLR. Thanks to EU/ESA/KNMI/DLR for providing the TROPOMI/S5P Level 1 products and NASA Worldview for the VIIRS/SNPP images used in this paper. We hereby acknowledge financial support from DLR (S5P KTR 2472046) for the development of TROPOMI retrieval algorithms.

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

**Table 1: Ranges for the input parameters appropriate for radiance simulations in the total ozone fitting window; ozone profiles are classified as a function of the total column. Smart sampling is employed to generate node points optimally covering all input dimensions and more than $2 \times 10^5$ synthetic UV spectra are generated.**

| Parameter | Minimun | Maximum |
|---|---|---|
| Ozone Profile | 125 DU | 575 DU |
| Solar Zenith Angle | 0° | 90° |
| Viewing Zenith Angle | 0° | 70° |
| Relative Azimuth Angle | 0° | 180° |
| Surface Albedo | 0 | 1 |
| Surface Pressure | 125 hPa | 1013 hPa |

**Table 2: Summary of the comparison between TROPOMI GE_LER clear-sky and OMI LER (first three rows) and between TROPOMI GE_LER cloudy and ROCINN_CRB cloud albedo (rows 4-6). There are more than 4.5 million clear-sky and more than 1.4 million cloudy cases out of approximately 15 million S5P measurements in April 10[th], 2018.**

|  | Number | Mean | Std. Dev. |
|---|---|---|---|
| Clear-sky Land | 866 907 | 0.0014 | 0.0624 |
| Clear-sky Water | 1 837 686 | -0.0144 | 0.0762 |
| Clear-sky Snow/Ice | 1 852 222 | -0.0048 | 0.2573 |
| Cloudy Land | 254 645 | 0.0834 | 0.1865 |
| Cloudy Water | 1 084 985 | 0.0487 | 0.1464 |
| Cloudy Snow/Ice | 127 636 | -0.0343 | 0.5432 |

**Table 3: Latitudinal differences between total ozone from CAMS and S5P using TROPOMI G3_LER and OMI LER for the month of April 2018. The values represent the total number of measurements for each latitudinal range and the mean differences ± standard deviations (in percentages). Latitude bands with less than 100000 data points were skipped, due to polar winter conditions, there are hardly any data south of 81°S. The number of measurements increases towards higher north because of overlapping orbits.**

| Latitude Range | Number | TROPOMI G3_LER | OMI LER |
|---|---|---|---|
| 80°S-70°S | 11297206 | -1.341 ± 2.364 | -2.041±2.114 |
| 70°S-60°S | 29018428 | -0.364 ± 2.472 | -2.727±2.300 |
| 60°S-50°S | 32351377 | 0.557 ± 1.783 | 0.808±1.815 |
| 50°S-40°S | 31580917 | -0.345 ± 1.189 | 0.048±1.224 |
| 40°S-30°S | 31154717 | -0.776 ± 0.906 | -0.336±0.930 |
| 30°S-20°S | 30948143 | -0.726 ± 0.770 | -0.252±0.807 |
| 20°S-10°S | 30814933 | -0.001 ± 0.736 | 0.537±0.745 |
| 10°S-0°S | 30744238 | -0.163 ± 0.774 | 0.517±0.720 |
| 0°N-10°N | 30732173 | -0.199 ± 0.833 | 0.607±0.738 |
| 10°N-20°N | 30779225 | -0.581 ± 0.798 | 0.142±0.728 |
| 20°N-30°N | 30894360 | -0.788 ± 0.945 | -0.097±0.901 |
| 30°N-40°N | 31091907 | -0.710 ± 1.340 | 0.173±1.336 |
| 40°N-50°N | 31469922 | -0.456 ± 1.858 | 0.584±1.880 |
| 50°N-60°N | 32250750 | -0.474 ± 1.721 | 1.287±1.920 |
| 60°N-70°N | 39590441 | -0.977 ± 2.211 | 1.155±2.798 |
| 70°N-80°N | 56545121 | -1.182 ± 2.581 | -0.730±2.701 |
| 80°N-90°N | 26178029 | -1.717 ± 2.424 | -1.595±2.317 |

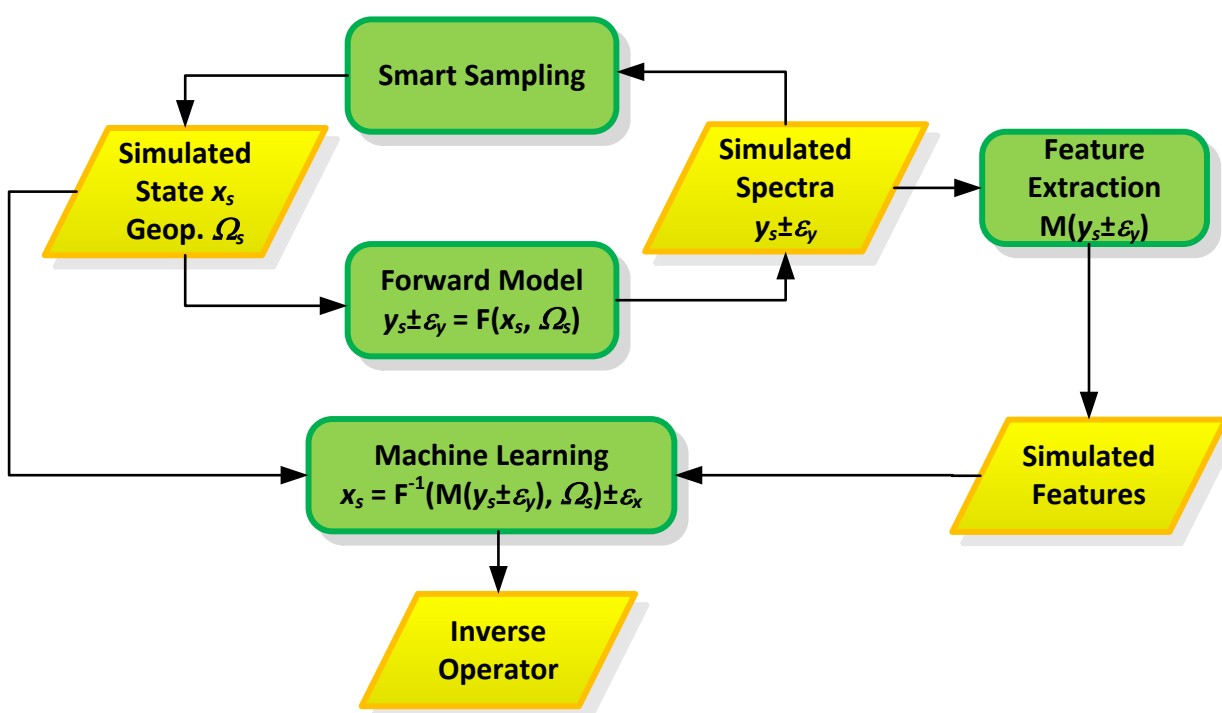

**Figure 1: Data flow diagram for the FP_ILM training phase. The smart sampling technique creates simulated state vector $x_s$ and geophysical conditions $\Omega_s$ that are used as input to the forward model for the calculation of simulated spectra with their expected errors $y_s + e_y$. Machine learning techniques are deployed for computing the inverse operator that is trained using as input the features extracted from the simulated spectra $M(y_s)$ and the geophysical conditions $\Omega_s$ as an output the state vector and the errors $x_s + e_x$.**

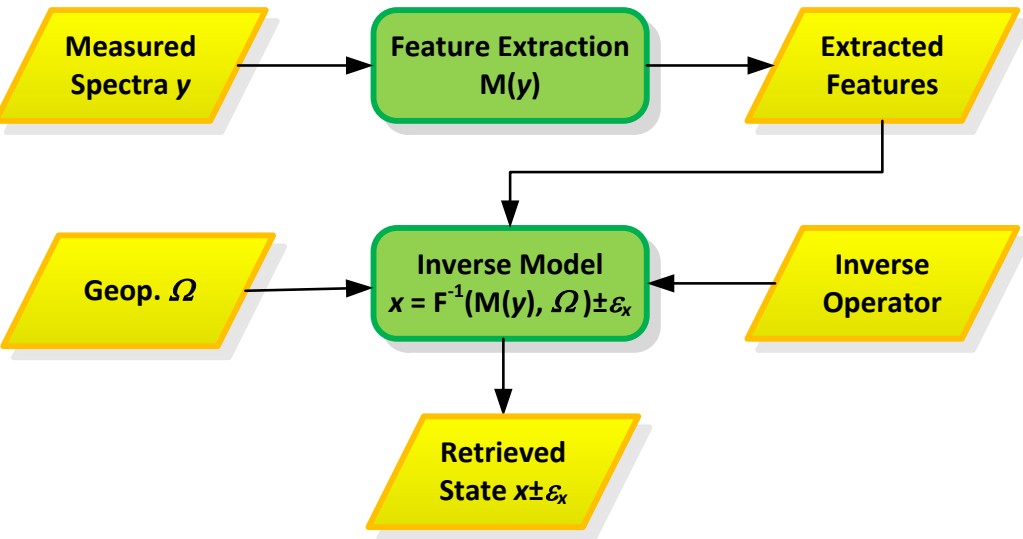

**Figure 2: Data flow diagram for the FP_ILM retrieval phase. The inverse operator computed during the FP_ILM training phase solves the inverse problem and retrieve the state vector $x$ taking as input the features $M(y)$ extracted from the measured spectra $y$ and geophysical conditions $\Omega$.**

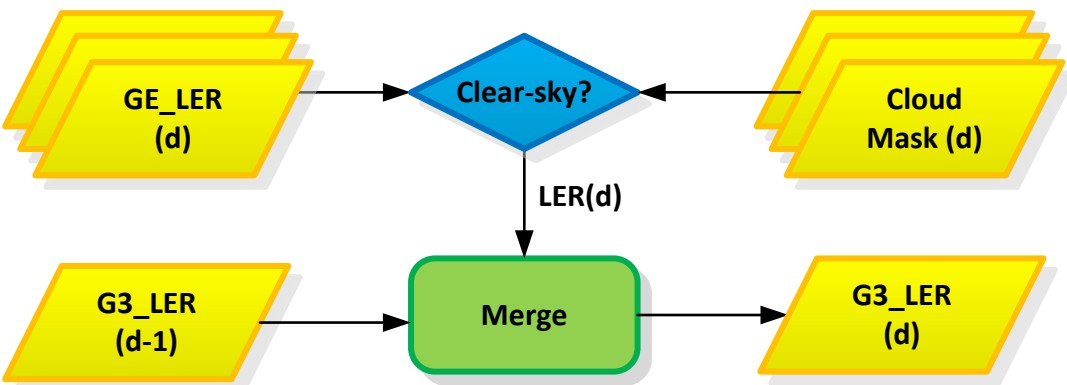

5  **Figure 3: Data flow diagram for the creation of the global gapless geometry-dependent LER (G3_LER) map for day *d*, obtained by merging the clear-sky LER data from the same day with the G3_LER map from the previous day.**

(a)  (b)  (c)

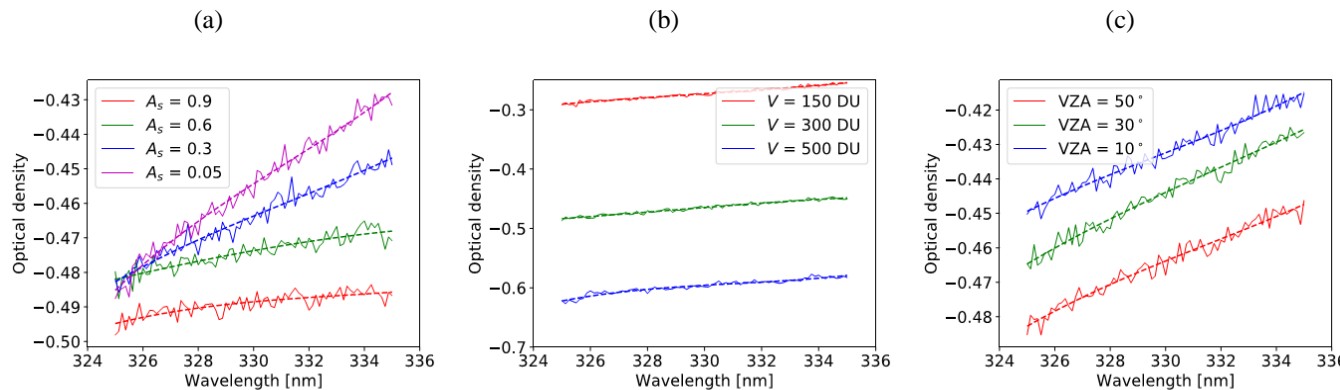

**Figure 4: Optical densities of the DOAS polynomial as a function of wavelength: with respect to (a) surface albedo, (b) total ozone, and (c) viewing zenith angle. The dotted-lines represent the DOAS fitted polynomial.**

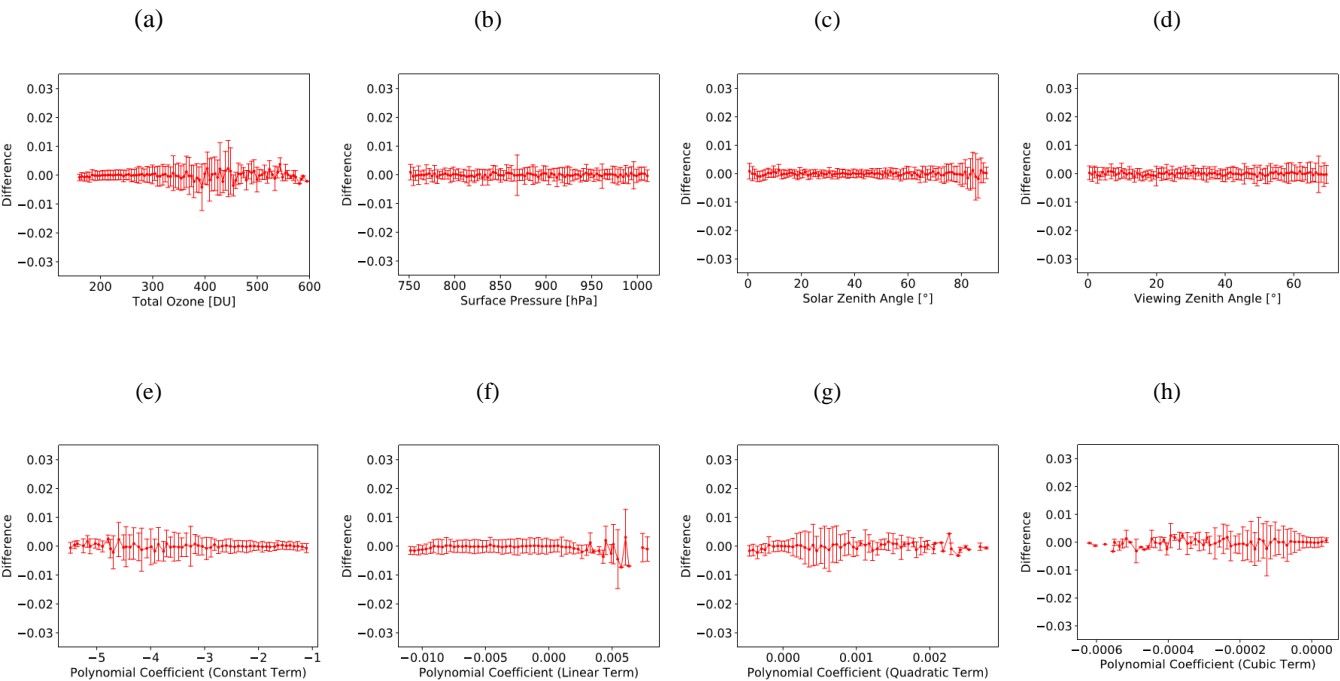

**Figure 5: GE_LER retrieval error as a function of (a) total ozone, (b) surface pressure, (c) solar zenith angle, (d) viewing zenith angle, and (e to h) the four DOAS polynomial fitting coefficients.**

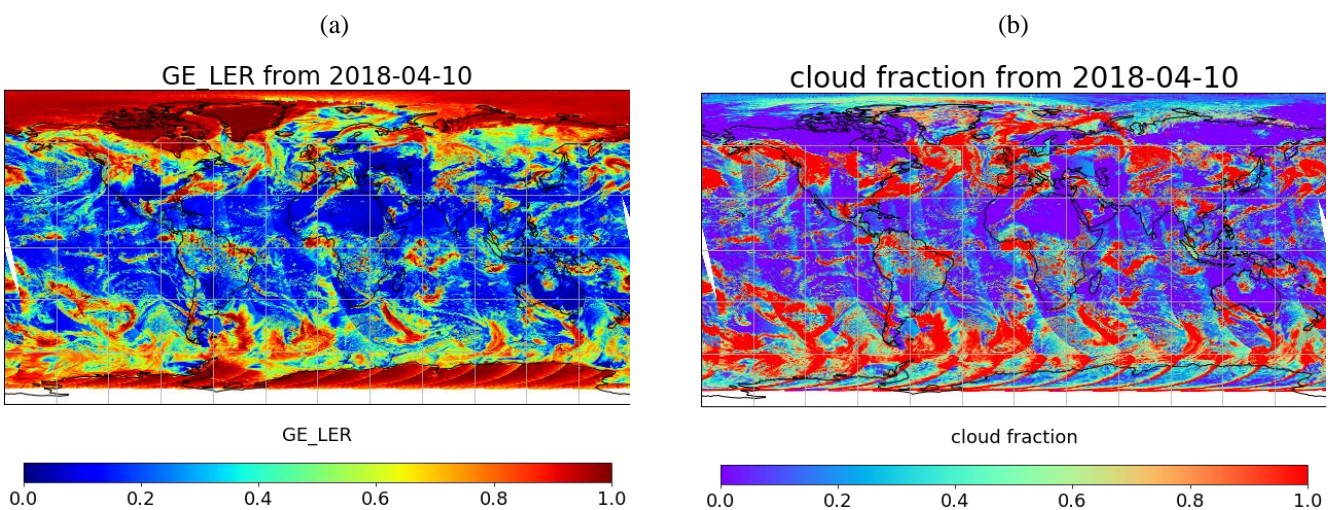

**Figure 6: (a) GE_LER in the total ozone fitting window [325-335 nm] retrieved from TROPOMI/S5P data on April 10<sup>th</sup>, 2018 and (b) the corresponding cloud fraction for this day.**

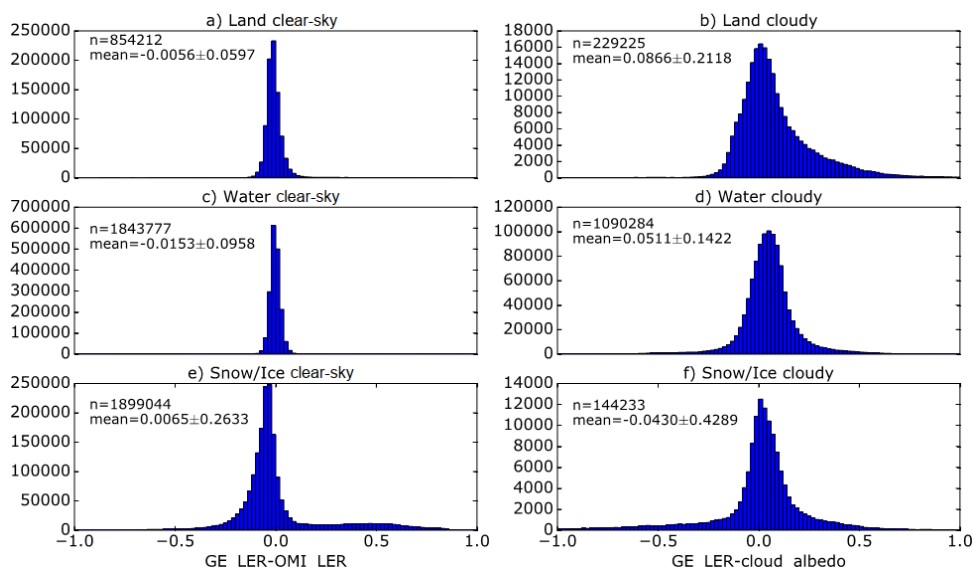

**Figure 7: Histograms of the differences (left) between clear-sky TROPOMI GE_LER and OMI LER climatology and (right) between the cloudy TROPOMI GE_LER and the ROCINN_CRB cloud albedo from the operational S5P cloud product. The comparisons are performed separately according to surface types (land, water, and snow/ice), with S5P data from April 10th 2018.**

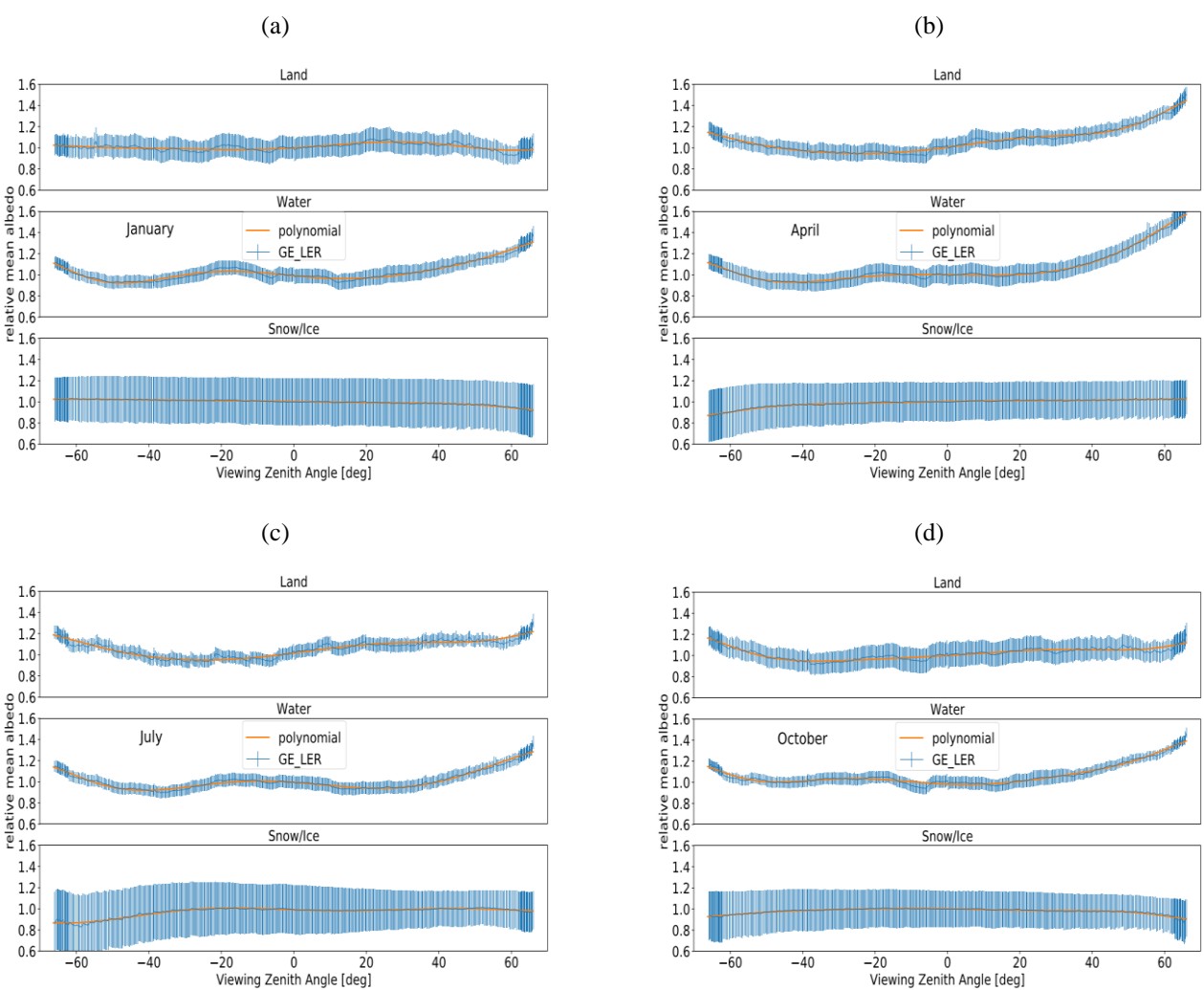

**Figure 8: BRDF dependencies ρ($\theta$) as a function of the viewing zenith angle for land, water, and snow/ice conditions, as calculated with normalized TROPOMI/S5P data from (a) January, (b) April, (c) July, and (d) October 2018. The negative viewing zenith angles correspond to the first 225 detector pixels.**

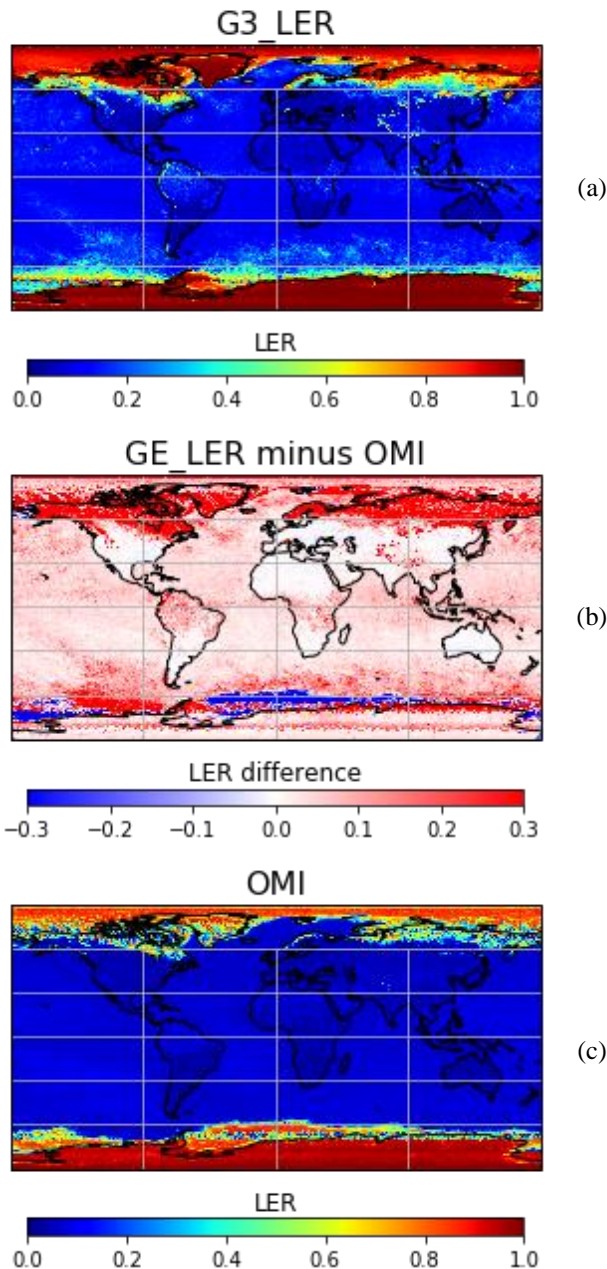

**Figure 9: (a) TROPOMI G3_LER daily map (325-335 nm) for April 30[th], 2018, (c) OMI LER climatology (335 nm) for the month of April, and (b) the difference between these two datasets. There is very good agreement over land and water surfaces, with major differences in snow/ice regions of the OMI LER climatology from 2004-2007 that do not match with actual surface conditions observed in April 2018.**

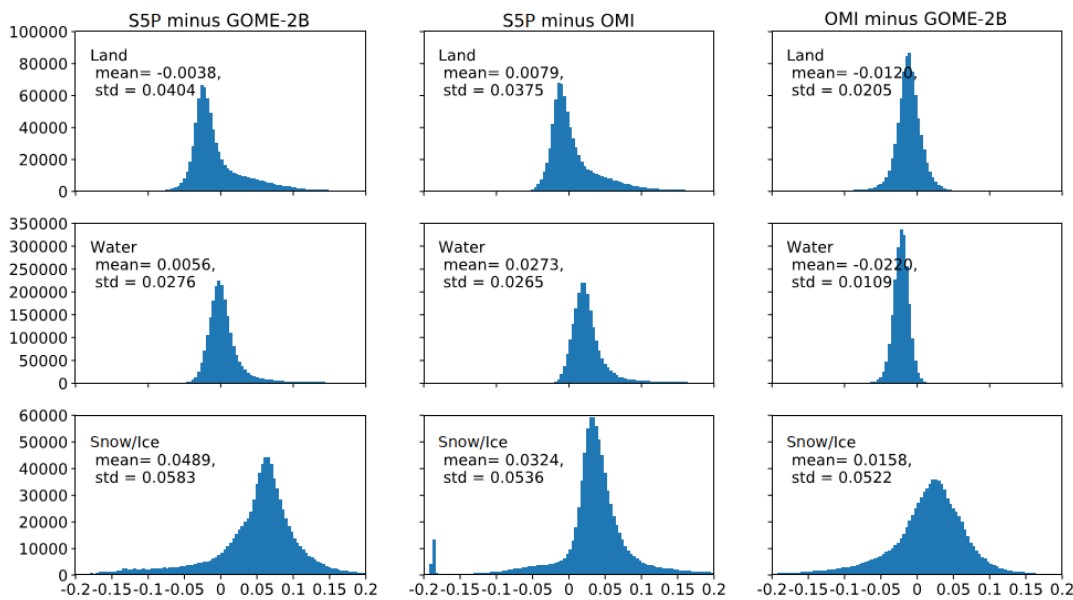

**Figure 10: Histograms of the differences (left) between TROPOMI G3_LER and GOME-2B climatology, (middle) between TROPOMI G3_LER and OMI LER climatology, and (right) between OMI and GOME-2B LER climatologies. The comparisons are performed separately for surface types (land, water, and snow/ice) using data from October 2018.**

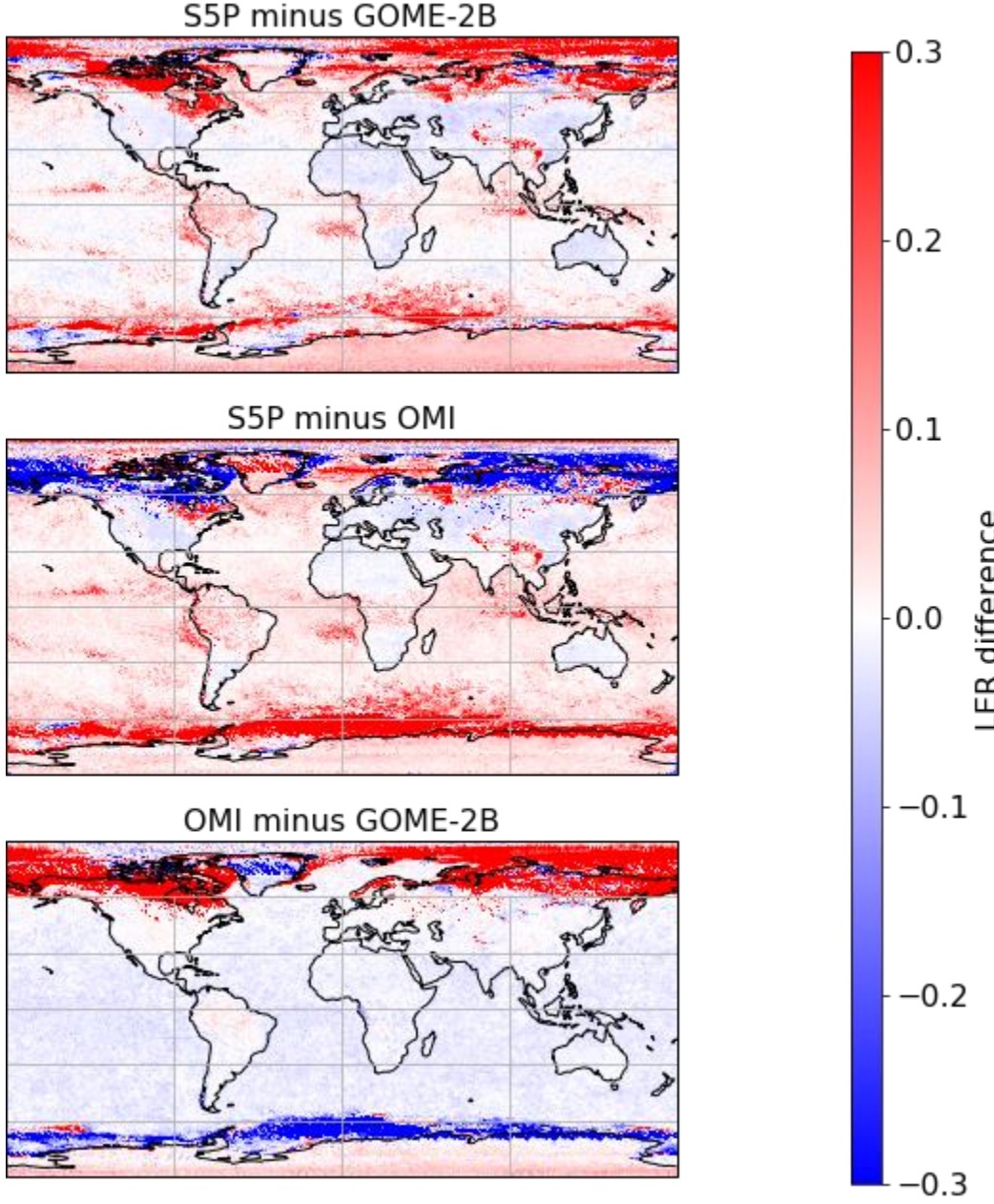

**Figure 11: Albedo difference maps between TROPOMI, GOME-2 and OMI for October 2018. North of 60°N the discrepancy between the three datasets reaches a maximum due to snow/ice conditions. While S5P overestimates compared to GOME-2, it underestimates compared to OMI.**

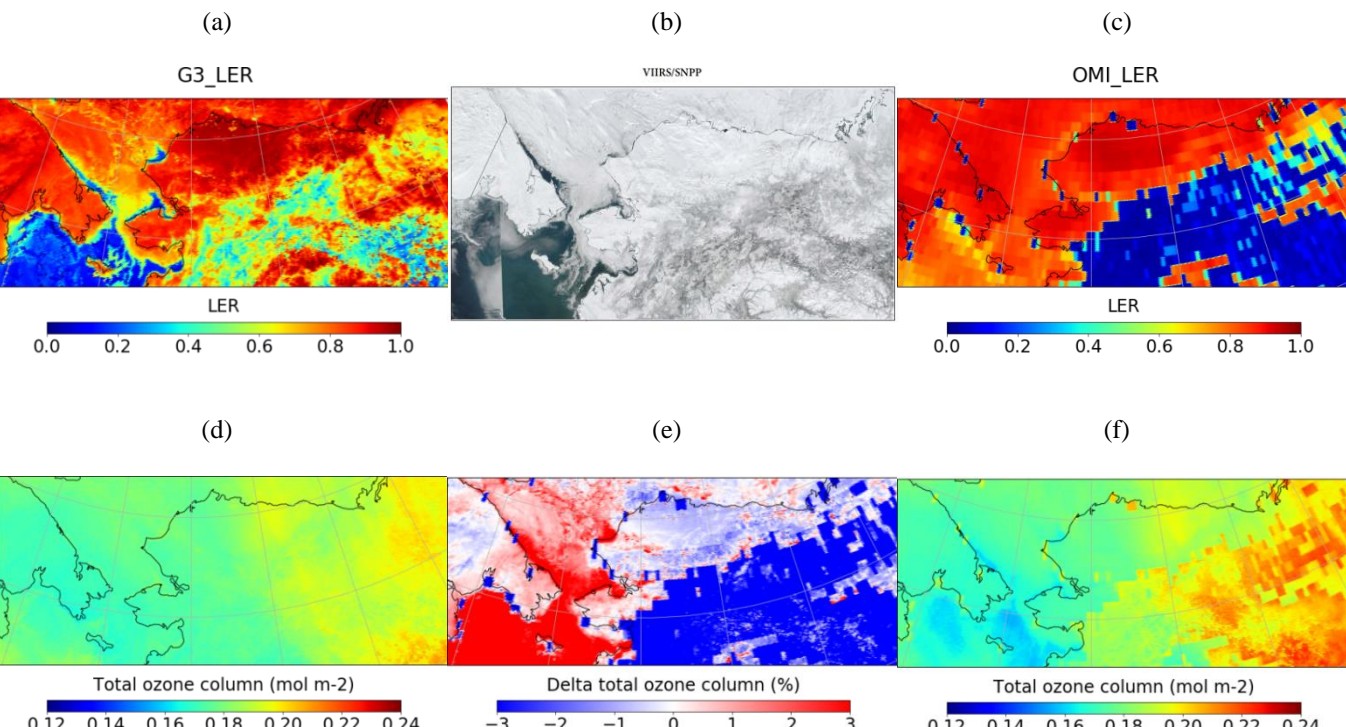

(a) G3_LER  (b) VIIRS/SNPP  (c) OMI_LER

LER
0.0  0.2  0.4  0.6  0.8  1.0

LER
0.0  0.2  0.4  0.6  0.8  1.0

(d)  (e)  (f)

Total ozone column (mol m-2)
0.12  0.14  0.16  0.18  0.20  0.22  0.24

Delta total ozone column (%)
−3  −2  −1  0  1  2  3

Total ozone column (mol m-2)
0.12  0.14  0.16  0.18  0.20  0.22  0.24

**Figure 12: TROPOMI/S5P (top) surface and (bottom) ozone measurements from April 1st, 2018 around the Bering Strait. The (a) TROPOMI/S5P G3_LER daily map agrees very well with the surface types observed in the (b) VIIRS/SNPP image coastal waters of Russia and Alaska. These coastal waters as well as the open waters of the Bering Sea are not properly represented in the (c) OMI LER climatology that shows snow/ice over these regions. Likewise, the OMI LER climatology erroneously shows no snow/ice in Alaska. The total ozone field using the (d) TROPOMI G3_LER daily map is significantly smoother than the field derived from the (f) OMI LER climatology. The coarse spatial resolution of the OMI LER climatology is clearly manifested in the total ozone field and incorrect snow/ice values in the OMI LER climatology induce large errors in the retrieved total ozone (e) with differences between −10% and +15%.**

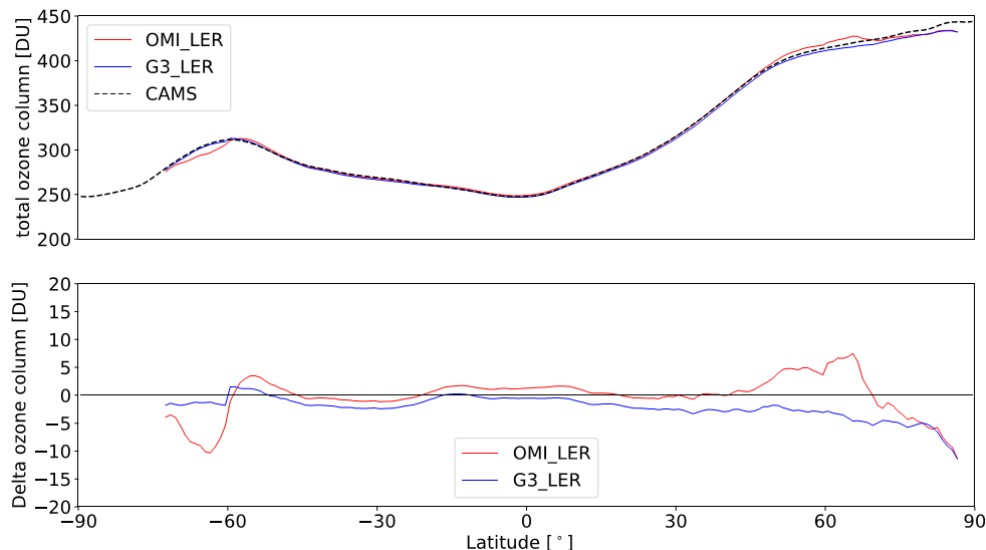

**Figure 13: Comparison of total ozone from CAMS and the S5P retrieved ozone using the OMI LER climatology and the daily TROPOMI G3_LER maps for April 2018. Total ozone values based on daily G3_LER maps is significantly closer to those from CAMS especially for high latitude regions.**

