# Peer review of "Applying FP\_ILM to the retrieval of geometry-dependent effective Lambertian equivalent reflectivity (GE\_LER) daily maps from UVN satellite measurements"

_Atmospheric Measurement Techniques, 2019_

## Referee Comment (RC1) · Anonymous Referee #1 · 27 May 2019

*Journal: AMT*
*Title: Applying FP_ILM to the retrieval of geometry-dependent effective Lambertian equivalent reflectivity (GE_LER) to account for BRDF effects on UVN satellite measurements of trace gases, clouds and aerosols*
*Author(s): Diego G. Loyola et al.*
*MS No.: amt-2019-37*

**General comments**

| # | page | line | comment |
|---|------|------|---------|
| 1 | | | The paper describes a novel method to derive the geometry dependent Lambert Equivalent Reflectance of the Earth scene, which is an important parameter needed for the retrieval of trace gases. The method is shown to have many benefits over the use of a climatology, as has been used often for past missions. The introduction of the paper is well written and of good quality, nonetheless, the remainder of the paper is a bit thin when it comes to provide evidence of the improvements over existing climatologies. Only comparisons with OMI are given while there exists more climatologies based on other missions. Also the directional aspect of the GE_LER needs more validation. The paper covers new and interesting topics and techniques, and after the comments (some of which major) and corrections have been adequately addressed, the paper could certainly be published. |
| 2 | | | Although the paper stresses the importance of the inclusion of BRDF in the newly derived TROPOMI surface reflectivity, this is not the only factor that plays a role, and probably not the strongest factor. Since the radiation field in the UV is largely diffuse, the actual BRDF of the surface is not so important. The better inclusion of snow/ice areas and the higher spatial resolution probably play a stronger role. Please discuss this point, and try to separate the effects of the three factors: BRDF, snow/ice, and spatial resolution in the comparison of TROPOMI GE-LER with OMI-LER climatology.

The improvement that is found in the TROPOMI total ozone retrieval in Fig. 11 when using the TROPOMI GE-LER instead of the OMI-LER is apparently due to the better snow/ice mapping at high latitudes, not to BRDF effects. |
| 3 | | | Are the GE_LER data available to the community? Please specify whether and how you plan to distribute the GE_LER and G3_LER data products. In order for other people to reproduce your results and claims they need open access to the data presented in this paper. |
| 4 | | | Which are the wavelengths for which GE_LER is retrieved? In the paper it is not so clear for which wavelength the results apply. For instance, only in the caption of Figure 6 this is mentioned. |
| | | | |

**Specific comments**

| # | page | line | comment |
|---|------|------|---------|
| 1 | 1 | | The title is hardly readable due to the many acronyms. Please make the title clearer. In the rest of the paper the construction "FP_ILM GE_LER" is hardly readable. Can you think of a better name? |
| 2 | 2 | 16 | These are not fundamental problems of a climatology itself, but rather information missing in the currently available climatologies. It would definitely be possible to create a climatology that includes the viewing angle dependency, or address separately snow and snow-free conditions. |
| 3 | 3 | 15 | The drawbacks mentioned for lookup tables are not very convincing, consider rephrasing this sentence. |
| 4 | 4 | 8 | The smart sampling technique should be explained in a bit more detail because readers may not want to read the full paper referred to. |
| 5 | 4 | 16 | I do not understand this sentence: "*Machine learning techniques perform best with low-dimensional datasets by avoiding the effects of the curse of dimensionality.*" |
| 6 | 5 | 27 | What about the azimuth dependence of \rho ? This also holds for other places in the paper. Please clarify in Sect. 2 how you deal with the solar zenith angle and relative azimuth dependence of the BRDF. |
| 7 | 7 | 9 | How did you calculate the standard deviation, is it the based on all simulations in the validation training set? Figure 5 on page 22 seems to indicate larger errors (up to 0.01) for individual LER retrievals. What are these red error bars in this figure? How does this error propagate in the final accuracy of the trace gasses? |
| 8 | 7 | 15 | Why do you use Z as symbol for pressure and not P? Z can easily be confused with height. |
| 9 | 7 | 21 | The histograms presented in Figure 7 are not discussed in detail. |
| 10 | 7 | | section 4.3 / Figure 9: This should become a separate main section, with a thorough and complete validation of the product. The comparison that is presented is not sufficient. Comparisons can be performed with a number of the surface LER -databases that were mentioned in the introduction (OMI, SCIAMACHY, GOME-2), but also with BRDF information from MODIS. Using MODIS BRDF would mean adjusting the retrieval to retrieve wavelengths of the nearest MODIS band. Can this be done? The differences have to be analysed properly. The difference plot in Figure 9(b) does not allow the reader to study differences on the order of 0.02, which is the typical difference/error one would expect for snow-free areas. |
| 11 | 8 | 2 | "*from the couple of days*": how many days did you use? |
| | 8 | 11 | Figure 8 needs more explanation, what order polynomial is used, what do the blue error bars represent? Why do land, water and snow scenes all have more or less the same relative albedo (around 1.0 – 1.6)? Have you calculated this average using all global pixels? This implies that you have mixed different land types in the calculation of the average. How representative is the viewing angle dependency then for individual land types? |
| 12 | 8 | 15 | Please check which version of the OMI LER was used; the second version covers 5 years of data between 2005 – 2009, released in 2010. |
| 13 | 8 | | Which field of the OMI-LER is used to compare with? Is it the |

| | | | |
|---|---|---|---|
| | | | "MonthlyMinimumSurfaceReflectance" field or is it the "MonthlySurfaceReflectance" field? |
| 14 | 11 ff | | References: please put all references in alphabetical order. |
| 15 | Fig. 5 | | Did you also consider the sensitivity of the GE_LER error due to ozone profile assumptions? |
| 16 | Fig. 8 | | What do you mean with *relative mean albedo*? Can you please also provide the GE_LER itself? |
| 17 | Fig. 9 | | These maps are not very informative because the dynamic range is too large. Please choose a color scale and albedo range that provides spatial information on the distribution of surface albedo in the UV. |
| | | | |

**Technical corrections**

| # | page | line | comment |
|---|------|------|---------|
| 1 | 1 | 21 | AMF abbreviation not explained |
| 2 | 2 | 1 | Typo mayor → major |
| 3 | 9 | 5 | Typo worst → worse, also in caption of figure 10 |
| 4 | all | | everywhere: "Lambertian-equivalent reflectivity" (with a "-") |
| 5 | 2 | 12 | page 2: "The unprecedented spatial resolution of TROPOMI..." → add "for a spectrometer" |
| | | | |

---

## Referee Comment (RC2) · Anonymous Referee #2 · 22 Jul 2019

This manuscript presents a new approach to derive effective scene albedo on a pixel-per-pixel basis from TROPOMI observations and to build a viewing zenith angle dependent LER climatology with an improved spatial resolution compared to former data bases. Although the topic of the study fits well within AMT and there is no obvious issue with the approach, I would suggest to further discuss the results and to extend the comparisons to better demonstrate the added-value of the database. For example, results are discussed for only one spectral region and a limited amount of data (April

2018). It would be beneficial to have more illustrations for different months. Reading the manuscript, I had many comments similar to those from reviewer 1. I won't list those again but encourage the authors to carefully reply to them. Below are a few additional comments. Once the comments have been addressed and the manuscript consolidated, this work will be worth being published within AMT.

**Comments:**

- The description of the smart sampling and machine learning approaches is quite technical. It would be beneficial to the readers to further describe the general ideas/concepts on which rely those methods.

- Section 3: Could you provide more details here on how clear-sky pixels are selected? Such details are given later in the manuscript but it would good to already describe this in section 3. Could you also provide some statistics on the number of days required to have a global coverage? There must be some regions with persistent clouds for which the update frequency drastically decreases. Actually, it would be useful for traceability to provide this information in the database along with the G3_LER values. For example, for one given cell, the LER value has been derived from day-1, -2, -3, or . . .

- G3_LER data seems to be available only for the ozone fitting window and for three surface types. Could you comment why only those three surface have been considered? In other regions than UV, BRDFs effects will differ much more significantly as a function of the surface type. Could you clarify if you intend to provide GLER data in other spectral ranges and how you intend to proceed with respect to this aspect.

- It is mentioned that the Bodeker ozone database is combined with the McPeters/Labow climatology as input of the RT simulations. Could you be more specific on the needs for this combination and on what is provided by each of
those databases. Also in Table 1, the ozone profiles appear to be classified only as a function of the total column. Is it sufficient or are the geographic variations of the profiles accounted for somehow? Is there any latitude/longitude dependence taken into account? If not, please be more specific on the profiles that have been used. Also could you provide typical sampling steps of the different dimensions?

- Figure 7: to better illustrate the possible impact of BRDF, could you show such clear-sky histograms for different range of viewing angles. If BRDF effect is important, we could expect systematic biases varying as a function of the VZA. Also, biases are more important for cloud cases. Is it because cloud albedo are retrieved in a different spectral region?

- Figure 8 : what are the implications of the numerical instability of the RT simulations around VZA=0 on the retrieved LER?

- Figure 9: There is a clear general bias between the G3_LER and OMI_LER data, even at low/mid-latitudes. Could you better quantify and discuss this? Is there any indication that one of the two data sets would be more realistic?

**Minor/Technical comments:**

- Quality of figures is generally low. Could you increase the quality as well as the size of labels?

- Page 2 – line 6: 35% on ozone column seems large. Is this value correct?

- Page 2 – line 12: Could you be more specific with that statement? Are there some references providing estimates of errors on TROPOMI products caused by the too coarse resolution of old databases?

- Page 3 – line 29: Add "solar and" before "viewing geometry"?

- Page 6 – line 8: The LER data could still differ from the actual surface properties in case of sudden snow fall combined with significant cloudiness.

- Page 6 – line 16: remove "viewing geometry"

- Figure 4 shows negative optical densities, which is not physical. In the text, those quantities are referred to optical densities differences but it is not clear what is the reference. Could you homogenize the text and y-label and clarify what are those optical density differences?

- Page 7 – lines 5-6: This is very technical and the meaning is not clear at all for me. Could you rephrase this?

- Page 8 – line 2: "from the couple of days" is not clear. Please be more specific.

- Page 8 – line 29: "smoother" instead of "smother"

- Page 9 – line 31: Mention that those numbers are valid for April 2018.

---

## Referee Comment (RC3) · Anonymous Referee #3 · 30 Jul 2019

Mateer et al. (1971) first proposed a Lambertian Equivalent Reflectivity (LER) concept in BUV total ozone retrievals to account for combined spectral dependence of surface, aerosols and cloud reflectance [Mateer, C. L.., D. F. Heath, A. J. Krueger: Estimation of Total Ozone from Satellite Measurements of Backscattered Ultraviolet Earth Radiance, J. Atmos. Sci., 28, 1307-1311, https://doi.org/10.1175/1520-0469(1971)028<1307:EOTOFS>2.0.CO;2, 1971]. The concept works well because ∼90% ozone is in the stratosphere, above effective reflecting surface. The simple LER

concept with some modifications (e.g. extrapolated LER spectral dependence, effective surface pressure) has been successfully used in heritage (TOMS, GOME, SCIA-MACHY) and current (GOME-2, OMI, OMPS, S5P/TROPOMI) stratospheric ozone and other trace gases (e.g., volcanic SO2) BUV retrievals.

The need for satellite retrievals of tropospheric ozone and other pollution gases (NO2, SO2, HCHO) in partly cloudy scenes, with peak concentrations in or just above the planetary boundary layer, required modification of the simple LER concept, replacing it with the mixed-LER (MLER) concept: mixing two LER surfaces, one at the ground and the other at the effective cloud pressure, e.g., [Ahmad, Z. et al: Spectral properties of backscattered UV radiation in cloudy atmospheres, J. Geophys. Res. Atmos., 109, D01201, https://doi.org/10.1029/2003JD003395, 2004; Stammes, P., et al.: Effective cloud fractions from the Ozone Monitoring Instrument: Theoretical framework and validation, J. Geophys. Res., 113, D16S38, https://doi.org/10.1029/2007JD008820, 2008]. The MLER approach is currently used in operational BUV pollution gas retrievals (e.g., [Levelt et al.: The Ozone Monitoring Instrument: overview of 14 years in space, Atmos. Chem. Phys., 18, 5699-5745, https://doi.org/10.5194/acp-18-5699-2018, 2018] and references therein). The MLER approach requires a-priori "clear-sky" LER estimate, which can be taken either from concurrent satellite measurements (e.g., OMI geometry-dependent GLER product uses higher-resolution atmospherically corrected MODIS BRDF [Vasilkov et al., AMT 2017]) or from prior measurements (e.g., OMI cloud-cleared climatological LER [Kleipool et al., JGR 2008]). The climatological "clear-sky" LER estimation is less accurate, since it disregards the observational geometry- and time-dependence of surface reflectance.

The paper by Loyola et al. presents new geometry-dependent (GE-LER) LER implementation, the "Full Physics – inverse Learning Machine (FP_ILM)" algorithm and the multiple day gridded LER product (G3_LER) derived from the present and previous clear-sky scenes observed by S5P/TROPOMI. In previous LER implementations for ozone retrievals, the LER values were derived at non-absorbing wavelengths (e.g.,

340nm and 380nm for Nimbus-7 TOMS) and spectrally interpolated to the ozone and SO2 retrieval windows. The important advantage of the new GE-LER retrieval is that it is retrieved in the same spectral fitting window used by ozone retrieval (325-335nm), thus does not require spectral extrapolation. This is the first simultaneous retrieval of both ozone and LER in this spectral window. The G3_LER can be applied to existing S5P aerosol, clouds and trace gas algorithms by replacing climatological clear-sky LER with the new G3_LER product.

I recommend publishing the paper with clarifications and technical corrections and releasing the new S5P GE_LER and gridded G3_LER products for community evaluation.

General comments

1) The name "full physics" is misleading, because the forward radiative transfer model used for NN training does not include important physical processes, such as , aerosols and inelastic (RRS) scattering;

2) acknowledge that BRDF effects on trace gas retrievals cannot be modeled exactly using forward RTM with Lambertian surface. Estimate the ozone errors due to Lambertian surface assumption (GE_LER or simple LER) using BRDF supplement available in VLIDORT RTM.

3) Provide more details about GE_LER algorithm:

a. Do you assume that GE_LER is wavelength independent within DOAS fitting window?

b. Give reference to the machine learning (NN) software and explain selecting optimal NN topology used in the algorithm training.

c. Clarify whether the RTM with Lambertian surface or with BRDF model was used for training?

[Figure]

d. Explain which cloud masking algorithm was used in creating G3_LER clear-sky daily map e. Fig. 1– clarify that "simulated features" are DOAS ozone slant columns and polynomial closure coefficients.

f. Fig. 2 – clarify that "extracted features" are DOAS ozone slant columns and polynomial closure coefficients.

4) Clarify what are effects of UV-absorbing aerosols (dust or smoke) on GE_LER?

5) Clarify that the neural network is trained on synthetic clear-sky spectra, but applied to the TROPKMI measurements over mixed, partly cloudy scenes (equation 5).

6) Compare TROPOMI GE_LER retrievals with the traditional LER retrievals at 340nm, where ozone absorption is negligible. Add TROPOMI simple LER340 map to Figure 10.

7) Publicly release G3_LER data set for community evaluation.

Technical comments

Table 2 is not mentioned in the text.

P1, 12: with a significant[ly] lower spatial resolution . . .

13: satellite viewing [geometry] dependencies

P2, 1: are mayor [major] error sources – clarify that the surface reflectance has larger influence on boundary layer trace gases retrievals and much less on the mid-and upper-tropospheric constituent retrievals.

13: significant[ly] lower spatial resolution

18: (b) the effect of surface reflectance anisotropy [is]

20: Retrieval of [Lambertian] effective scene albedo has been used in total ozone algorithms from nadir and limb – add pioneering reference: Mateer, C. L.., D. F. Heath, A. J. Krueger: Estimation of Total Ozone from Satellite Measurements of Backscattered Ultraviolet Earth Radiance, J. Atmos. Sci., 28, 1307-1311, https://doi.org/10.1175/1520-0469(1971)028<1307:EOTOFS>2.0.CO;2, 1971.

22: - add references to heritage TOMS ozone, e.g., Bhartia, P. K., et al.: Algorithm for the estimation of vertical ozone profiles from the backscattered ultraviolet technique, J. Geophys.Res., 101, 18793–718806, 1996 McPeters, et al.: Earth Probe Total Ozone Mapping Spectrometer (TOMS) Data Products User's Guide, NASA/TP-1998-206895, 1998.

- and OMI ozone references, e.g., McPeters, R. D., Frith, S., and Labow, G. J.: OMI total column ozone: extending the long-term data record, Atmos. Meas. Tech.,8, 4845–4850, https://doi.org/10.5194/amt-8-4845-2015, 2015.

Veefkind, J. P., et al.: Total ozone from the Ozone Monitoring Instrument (OMI) using the OMI-DOAS technique, IEEE T. Geosci. Remote, 44,1239–1244, 2006.

24: from other [higher spatial resolution] satellite sensors

28:" needed for computing LER from [and] BRDF may not be fully compatible" – need clarification: In Vasilkov et al., [2017] LER is calculated from the RT model simulated TOA radiance in a standard way, which is fully compatible with OMI cloud and NO2 retrievals. However, MODIS BRDF product may use different RT assumptions.

P3,

16: errors could be large and [multi-dimensional interpolations are] time consuming.

21: During the last years we [Recently] we developed an approached called . . .

22: applied for retrieving [ozone] profile shapes . . .

P4,

4, . . . the surface properties - clarify what properties? Did you use RTM with Lambertian surface for training or did you use RTM with BRDF model? Specify, which land/ocean BRDF model/dataset was used for training ?

resolution to resolve [absorbing] features usually contains [hyperspectral] radiances at a high-dimensional space

. . .avoiding the effects of the curse of dimensionality ? – clarify

Explain where does the GE_LER information comes from (i.e., equation (3) )?

P5, 19 . . . effective scene approximation - add reference ([Mateer et al., 1971; ] Coldewey-Egbers et al., 2005)

whereas a [clear-sky] LER is needed

GE_LER retrieved under clear sky conditions – explain cloud masking algorithm

24, Fig 3 . . . based on the [GE_]LER data from previous days – Clarify if the GE-LER map instrument and viewing geometry specific?

25-26 (BRDF) effects, as it is based on radiative transfer model simulations using the actual viewing geometry – clarify did you use RTM with Lambertian or BRDF surface? What surface BRDF model/dataset (if any) was used in creating training spectral dataset?

P6, fitting a polynomial of clear-sky LERs averaged as function of ðÌIJČ . – Please, clarify: - should BRDF function also depend on solar and azimuthal angles in addition to satellite view angle?

- Provide examples (add figure) of the clear sky LER(theta) for land and water surfaces.

synthetic UV spectra – clarify that spectra were simulated assuming Lambertian surface, no aerosols and no inelastic RRS effects.

**AMTD**

ozone [profile?] climatology

Figure 4 shows the optical densities difference – clarify definition of the optical density and the OD difference. Explain why is Figure 4 necessary?

... albedo of 0.05, 0.3, 0.6, and 0.9 [,which] correspond to water,.. – not clear how [ozone?] optical density is related to the surface albedo?

higher [longer?] wavelength.

P7,

1, Fig 4 . . . optical density increases when the viewing zenith angle decreases – please, explain. The ozone optical density is proportional to the slant column ozone amount, which should decrease when the viewing zenith angle decreases. . . .for all cases, the optical density increases along the wavelength region –Explain why is this important?

. . .is reorganized according to (3) – clarify the meaning of equation (3) an reorganization algorithm

. . . using a NN with a topology of 9-20-8-2-1, - provide reference to the NN software used and how the optimal topology has been selected?

10, Fig.5 . . .represents the inverse function [of the synthetic dataset] in a very precise way – this does not guarantee similar accuracy when applied to the real satellite measurements.

Figure 6(a) title and color bar show "E_LER" – change to GE_LER

Figure 6(b) – explain cloud fraction stripes over Antarctica?

In the case of clear-sky (ðİŚŞðİŚŘ $\leq$ 0.05 ) the GE_LER represents the surface albedo – clarify if GE_LER represents hemispherical albedo or directional BRF ?

the TROPOMI clear-sky GE_LER and OMI LER climatology – Add comparison with the OMI/TROPOMI simple LER at 340nm in Table 2.

[Figure]

summarized in Figure 7. - in Table 2? P8,

. . . aggregating normalized [GE_]LER from the couple of days. – these retrievals are obtained under different viewing geometries. - Couple of days may not be sufficient to obtain cloud-free observations over certain locations. - Explain how GE_LER are normalized and what viewing geometry does the aggregated G3_LER map correspond to?

. . . averaged as function of the viewing zenith angle. – BRDF depends also on solar zenith and relative solar azimuthal angles. Why is this dependence ignored?

Fig. 8 Why is sun-glint is not visible for the water surface GE_LER and "hot spot" is not visible for the land GE_LER ?

What would GE_LER look like for a cloud-free sun-glint region?

Fig. 9(a) – what viewing geometry does the aggregated G3_LER map corresponds to? Reduce upper scale or use logarithmic scale to better show LER variability for snow-free regions. Clarify wavelength for the OMI climatological LER.

Fig 9 caption: the ma[j]or differences

Fig 10. Add comparison with the TROPOMI simple LER map at 340nm (negligible ozone absorption)

associated to [with] the coarse resolution most important[ly]

p9, what is even wors[e]

reduced from $-2.53 \pm 2.46\%$ using OMI LER to $0.78 \pm 3.49\%$ using TROPOMI G3_LER - why did the standard deviation increase?

P12,

Loyola, D., et al.: The near-real-time total ozone retrieval algorithm from TROPOMI onboard Sentinel-5 Precursor, Atmos.Meas. Tech. Discuss., in preparation, 2019. – provide complete citation

---

## Author Response (AR1)

**Reply (in blue) to Referee #1**

We thank the referee #1 for the positive assessment of the paper. *Our reply is included after the referee comments.*

**1.** The paper describes a novel method to derive the geometry dependent Lambert Equivalent Reflectance of the Earth scene, which is an important parameter needed for the retrieval of trace gases. The method is shown to have many benefits over the use of a climatology, as has been used often for past missions. The introduction of the paper is well written and of good quality, nonetheless, the remainder of the paper is a bit thin when it comes to provide evidence of the improvements over existing climatologies. Only comparisons with OMI are given while there exists more climatologies based on other missions. Also the directional aspect of the GE\_LER needs more validation. The paper covers new and interesting topics and techniques, and after the comments (some of which major) and corrections have been adequately addressed, the paper could certainly be published.

In the updated paper (new Section 4.4) we include comparisons with OMI and GOME-2 LER

**2.** Although the paper stresses the importance of the inclusion of BRDF in the newly derived TROPOMI surface reflectivity, this is not the only factor that plays a role, and probably not the strongest factor. Since the radiation field in the UV is largely diffuse, the actual BRDF of the surface is not so important. The better inclusion of snow/ice areas and the higher spatial resolution probably play a stronger role. Please discuss this point, and try to separate the effects of the three factors: BRDF, snow/ice, and spatial resolution in the comparison of TROPOMI GE-LER with OMI-LER climatology. The improvement that is found in the TROPOMI total ozone retrieval in Fig. 11 when using the TROPOMI GE-LER instead of the OMI-LER is apparently due to the better snow/ice mapping at high latitudes, not to BRDF effects.

The reviewer is certainly right; in the UV the main improvements are from the accurate snow/ice retrievals whereas in the VIS the BRDF effects are stronger.

To better balance the different benefits from GE\_LER we have:

- Remove BRDF from the title, the new title is "Applying FP\_ILM to the retrieval of geometry-dependent effective Lambertian equivalent reflectivity (GE\_LER) daily maps from UVN satellite measurements"
- Emphasize in the introduction and conclusions the advantages of retrieving daily surface properties (especially important for snow/ice conditions) with the same spatial resolution and the same fitting window as the trace gases.

**3.** Are the GE\_LER data available to the community? Please specify whether and how you plan to distribute the GE\_LER and G3\_LER data products. In order for other people to reproduce your results and claims they need open access to the data presented in this paper.

The retrieved GE\_LER and the G3\_LER used for each single TROPOMI ground pixel will be included in the operational S5P total ozone product. All operational S5P products are open and free available. We will discuss with ESA/EU the possibility of disseminating the G3\_LER total ozone daily maps in the same way as the operational S5P products.

**4.** Which are the wavelengths for which GE\_LER is retrieved? In the paper it is not so clear for which wavelength the results apply. For instance, only in the caption of Figure 6 this is mentioned.

As mentioned in the Introduction and Conclusions, the GE\_LER/G3\_LER algorithms can be applied to any wavelength region. The examples shown for S5P are for the total ozone fitting window and the corresponding wavelengths are given in the first sentence of section 4 "...using the fitting window of 325-335 nm".

Additionally we added the fitting window information for the S5P examples in Section 4.2 and in the Conclusions.

**Specific comments**

**1 p1** The title is hardly readable due to the many acronyms. Please make the title clearer. In the rest of the paper the construction "FP\_ILM GE\_LER" is hardly readable. Can you think of a better name?

We simplified the title by removing the BRDF part. See the reply to the general comment#2

FP\_ILM GE\_LER together is indeed hard to read; in the updated paper we use only GE\_LER.

**2 p2 l16** These are not fundamental problems of a climatology itself, but rather information missing in the currently available climatologies. It would definitely be possible to create a climatology that includes the viewing angle dependency, or address separately snow and snow-free conditions.

That is correct, the sentence is reformulated to "common problems with typical LER climatologies"

**3 p3 l15** The drawbacks mentioned for lookup tables are not very convincing, consider rephrasing this sentence.

This sentence reads now "The main drawbacks of look-up tables representing high dimensional RTM simulations (common in atmospheric composition retrievals) are that the memory requirements increase exponentially with the number of input dimensions, the interpolation/extrapolation in this multi-dimensional space are computational expensive, and the interpolation/extrapolation errors could be significant."

**4 p4 l8** The smart sampling technique should be explained in a bit more detail because readers may not want to read the full paper referred to.

The following text is included in Section 2.2. "Training data is traditionally created at fixed intervals uniformly distributed for each input variable; as a consequence the training samples are grouped around the node points and a very poor coverage of the multidimensional input space is reached. Deterministic sampling methods provide a more uniform distribution of the training data covering the entire space of each input variable" and "For this work we select a Halton sequence that uses prime numbers for creating sample points in each input dimension and a radiative transfer model computes the corresponding simulated radiances".

**5 p4 l16** I do not understand this sentence: "Machine learning techniques perform best with low-dimensional datasets by avoiding the effects of the curse of dimensionality."

This second part of the sentence "by avoiding the effects of the curse of dimensionality" is removed.

**6 p5 l27** What about the azimuth dependence of rho? This also holds for other places in the paper. Please clarify in Sect. 2 how you deal with the solar zenith angle and relative azimuth dependence of the BRDF.

The following clarification is included in Section 3 "solar zenith angel dependencies can be ignored when combining GE\_LER data from Sun-synchronous satellites over the same position because the angle of sunlight upon the Earth's surface is consistently maintained. Likewise relative azimuth angle dependencies are negligible in the UV.".

**7 p7 l9** How did you calculate the standard deviation, is it the based on all simulations in the validation training set? Figure 5 on page 22 seems to indicate larger errors (up to 0.01) for individual LER retrievals. What are these red error bars in this figure? How does this error propagate in the final accuracy of the trace gasses?

Correct, we use all simulations in the validation dataset.

The following clarification is included in section 4.1 "the x-axes are divided in bins and the mean and standard deviation (red bars) are calculated for each bin."

The larger errors correspond to high SZA. The effects of LER errors on the trace gasses accuracy is discussed in the first sentence of the Introduction.

**8 p7 115** Why do you use Z as symbol for pressure and not P? Z can easily be confused with height.

Thanks for pointing out this inconsistency. The retrieval is actually based on surface height and not pressure. The symbol Z is correct, the text in Section 2 and 4 is updated.

9 p7 l21 The histograms presented in Figure 7 are not discussed in detail.

In chapter 4 we include a new section describing the comparison with GOME-2 and OMI LER.

**10 p7 section 4.3 / Figure 9:** This should become a separate main section, with a thorough and complete validation of the product. The comparison that is presented is not sufficient. Comparisons can be performed with a number of the surface LER -databases that were mentioned in the introduction (OMI, SCIAMACHY, GOME-2), but also with BRDF information from MODIS. Using MODIS BRDF would mean adjusting the retrieval to retrieve wavelengths of the nearest MODIS band. Can this be done?

The main focus of this paper is to present the algorithms for obtaining G3\_LER and G3\_LER, the results for S5P total ozone are shown as demonstration.

In chapter 4 we include a new section describing the comparison with GOME-2 and OMI LER.

The MODIS BRDF is available only in the VIS. As explained in the Conclusions, GE\_LER retrievals in the VIS are planned for future work.

The differences have to be analysed properly. The difference plot in Figure 9(b) does not allow the reader to study differences on the order of 0.02, which is the typical difference/error one would expect for snow-free areas.

In chapter 4 we include a new section describing the comparison with GOME-2 and OMI LER.

11 p8 l2 "from the couple of days": how many days did you use?

This sentence is reformulated as follows "The TROPOMI G3\_LER map for a given day is created by regridding (using a  $0.1^{\circ} \times 0.1^{\circ}$  resolution) the clear-sky LER data from the same day with the G3\_LER map based on the LER data from previous days"

**11 p8 l11** Figure 8 needs more explanation, what order polynomial is used, what do the blue error bars represent? Why do land, water and snow scenes all have more or less the same relative albedo (around 1.0 - 1.6)? Have you calculated this average using all global pixels?

This implies that you have mixed different land types in the calculation of the average. How representative is the viewing angle dependency then for individual land types?

The data for each surface type are normalised relative to the central detector pixel (nadir) therefore the range is around 1. This explanation is included in Section 4.3 and Figure 8.

**12 p8 l15** Please check which version of the OMI LER was used; the second version covers 5 years of data between 2005 – 2009, released in 2010.

The data used here are the 4 years data released in 2008. In an early stage of the S5P project we compared both the 2008 and the 2010 versions and found some systematic structures in the 2010 version especially in the 328 nm dataset. Therefore we decided to use 2008 version.

**13 p8** Which field of the OMI-LER is used to compare with? Is it the "MonthlyMinimumSurfaceReflectance" field or is it the "MonthlySurfaceReflectance" field?

We use the MonthlyMinimumSurfaceReflectance field.

14 p11ff References: please put all references in alphabetical order.

Done

**15 Fig 5** Did you also consider the sensitivity of the GE\_LER error due to ozone profile assumptions?

We are using ozone profile climatologies organized as function of the total ozone (the total ozone and the ozone profile are strongly correlated). Therefore the sensitivity of the GE\_LER to the ozone profiles is covered by the total ozone dependency.

**16 Fig 8** What do you mean with "relative mean albedo"? Can you please also provide the GE\_LER itself?

Please clarify, "relative mean albedo" is not mentioned in Fig. 8.

**17 Fig 19** These maps are not very informative because the dynamic range is too large. Please choose a color scale and albedo range that provides spatial information on the distribution of surface albedo in the UV.

Maps updated

**Reply (in blue) to Referee #2**

**We thank the referee #2 for the constructive comments.**

Our reply is included after the referee comments.

This manuscript presents a new approach to derive effective scene albedo on a pixel per-pixel basis from TROPOMI observations and to build a viewing zenith angle dependent LER climatology with an improved spatial resolution compared to former data bases. Although the topic of the study fits well within AMT and there is no obvious issue with the approach, I would suggest to further discuss the results and to extend the comparisons to better demonstrate the added-value of the database. For example, results are discussed for only one spectral region and a limited amount of data (April 2018).

It would be beneficial to have more illustrations for different months. Reading the manuscript, I had many comments similar to those from reviewer 1. I won't list those again but encourage the authors to carefully reply to them. Below are a few additional comments. Once the comments have been addressed and the manuscript

consolidated, this work will be worth being published within AMT.

As general reply we would like to highlight that the G3\_LER is not a climatology or database as commonly created by other methods but a dynamic map updated every day and in this way it represents the current surface conditions.

Furthermore, the main focus of this paper is to present the algorithms for obtaining G3\_LER and G3\_LER, the results for S5P UV (ozone) are shown as demonstration. Results for different seasons were already included in the submitted version, see for example Fig. 8.

**Comments:**

• The description of the smart sampling and machine learning approaches is quite technical. It would be beneficial to the readers to further describe the general ideas/concepts on which rely those methods.

The following text is included in Section 2.2. "Training data is traditionally created at fixed intervals uniformly distributed for each input variable; as a consequence the training samples are grouped around the node points and a very poor coverage of the multidimensional input space is reached. Deterministic sampling methods provide a more uniform distribution of the training data covering the entire space of each input variable" and "For this work we select a Halton sequence that uses prime numbers for creating sample points in each input dimension and a radiative transfer model computes the corresponding simulated radiances".

• Section 3: Could you provide more details here on how clear-sky pixels are selected? Such details are given later in the manuscript but it would good to already describe this in section 3. Could you also provide some statistics on the number of days required to have a global

coverage? There must be some regions with persistent clouds for which the update frequency drastically decreases. Actually, it would be useful for traceability to provide this information in the database along with the G3\_LER values. For example, for one given cell, the LER value has been derived from day-1, -2, -3, or . . .

TROPOMI and VIIRS data are used for the clear-sky determination. The following sentence is included in the paper: "In the case of S5P, clear-sky is determined using both the operational cloud properties retrieved from TROPOMI (Loyola et al., 2018) and the VIIRS/SNPP (Visible Infrared Imaging Radiometer Suite sensor onboard Suomi National Polar-orbiting Partnership satellite) cloud mask regridded to the TROPOMI resolution (R. Siddans, 2016). Note that S5P and SNPP fly in loose formation, the S5P orbit trails 3.5 to 5 minutes behind SNPP"

One month of data is usually enough for obtaining a globe map. The following explanation is added in section 3: "The spatial resolution of the G3\_LER maps for TROPOMI is 0.1° and global maps can be generally derived combining data from one month. Two to three months of data are only needed for regions covered with persistent clouds like the Intertropical Convergence Zone (ITCZ)."

The G3\_LER is not a classical static database, as explained in section 4.3 the G3\_LER maps are updated on a daily basis to represent the current surface conditions. To address the traceability question of the reviewer, we added the following in section 4.3: *"Time information (orbit number) of the LER used in each grid cell is included in the GE\_LER maps."*

• G3\_LER data seems to be available only for the ozone fitting window and for three surface types. Could you comment why only those three surface have been considered? In other regions than UV, BRDFs effects will differ much more significantly as a function of the surface type. Could you clarify if you intend to provide GLER data in other spectral ranges and how you intend to proceed with respect to this aspect.

The selected land, water and snow/ice cover well the BRDF effects in the UV. We include the following sentence in section 4.3 "Note that the selected surface types cover the BRDF effects in the UV ozone fitting window; other trace gases like NO2 in the VIS will require different land cover types (e.g. water, snow/ice, urban, paddy, crop, deciduous forest, evergreen forest) to properly model the BRDF effects, see Noguchi et al., 2014.".

Regarding the second question about the spectral ranges, in the Conclusions we already indicate our plans to apply the GE\_LER/G3\_LER to other S5P fitting windows.

• It is mentioned that the Bodeker ozone database is combined with the McPeters/Labow climatology as input of the RT simulations. Could you be more specific on the needs for this combination and on what is provided by each of those databases. Also in Table 1, the ozone profiles appear to be classified only as a function of the total column. Is it sufficient or are the geographic variations of the profiles accounted for somehow? Is there any latitude/longitude

dependence taken into account? If not, please be more specific on the profiles that have been used. Also could you provide typical sampling steps of the different dimensions?

The corresponding paragraph in section 4.1 was rephrased as follows: "We use the Bodeker et al., (2013) database for representing the stratospheric ozone combined with the McPeters/Labow (Labow et al., 2015) climatology for the lower tropospheric ozone."

A classification as function of the total column is sufficient thanks to the strong correlation between the total ozone and the ozone profile shape.

The smart sampling does not use "sampling steps". Please see our reply to your first comment.

• Figure 7: to better illustrate the possible impact of BRDF, could you show such clear-sky histograms for different range of viewing angles. If BRDF effect is important, we could expect systematic biases varying as a function of the VZA.

Also, biases are more important for cloud cases. Is it because cloud albedo are retrieved in a different spectral region?

We created histograms as function of VZA but they are not really informative. The BRDF dependency on the VZA can be better appreciated in the plots from Figure 8.

• Figure 8 : what are the implications of the numerical instability of the RT simulations around VZA=0 on the retrieved LER?

We removed the numerical instability and created extra simulations around nadir (VZA=0).

• Figure 9: There is a clear general bias between the G3\_LER and OMI\_LER data, even at low/mid-latitudes. Could you better quantify and discuss this? Is there any indication that one of the two data sets would be more realistic?

We found out that the small bias was due to imperfections on the current TROPOMI L1 products. The following sentence was added in section 4.2 "*The version 1 of the TROPOMI Level 1 product has small deficiencies on the UV band; therefore a soft-correction based on comparisons with OMPS radiances is applied to S5P. It is expected that the version 2 of the TROPOMI Level 1 product will include a more accurate radiometric calibration*".

We include a new section 4.4 describing the comparison with GOME-2 and OMI LER. However, it is not obvious which of the three data basis is actually best / most realistic. For some cases two of them agree well for other cases other two agree better.

**Minor/Technical comments:**

• Quality of figures is generally low. Could you increase the quality as well as the size of labels?

Figure quality and font size improved

• Page 2 – line 6: 35% on ozone column seems large. Is this value correct?

Figure 4 of Lerot et al. shows AMF changes in this magnitude when a surface albedo of snow/ice is used instead of surface albedo of water.

• Page 2 – line 12: Could you be more specific with that statement? Are there some references providing estimates of errors on TROPOMI products caused by the too coarse resolution of old databases?

Error estimates are not yet available, but this is a known data quality issue listed in the "Product Readme File" of the S5P L2 products, see http://www.tropomi.eu/documents/prf

• Page 3 – line 29: Add "solar and" before "viewing geometry"?

Done

• Page 6 – line 8: The LER data could still differ from the actual surface properties in case of sudden snow fall combined with significant cloudiness.

That is correct, at the end of the sentence we added *"The only exceptions are cases of sudden snow fall combined with significant cloudiness"*.

• Page 6 – line 16: remove "viewing geometry"

Done

• Figure 4 shows negative optical densities, which is not physical. In the text, those quantities are referred to optical densities differences but it is not clear what is the reference. Could you homogenize the text and y-label and clarify what are those optical density differences?

Negative optical densities are indeed misleading. What is shown here is the optical density of the DOAS polynomial ( $p(\lambda)$  in Equation 2). This information is added in Section 4.1 and in Figure 4.

• Page 7 – lines 5-6: This is very technical and the meaning is not clear at all for me. Could you rephrase this?

We use common nomenclature of machine learning; the text has been updated as follows: *"The best results are obtained using a feedforward NN (the neurons are grouped in layers) with a topology ... "*

• Page 8 – line 2: "from the couple of days" is not clear. Please be more specific.

This sentence is reformulated as follows "*The TROPOMI G3\_LER map for a given day is created by regridding (using a*  $0.1^{\circ} \times 0.1^{\circ}$  *resolution) the clear-sky LER data from the same day with the G3\_LER map based on the LER data from previous days*"

• Page 8 – line 29: "smoother" instead of "smother"

**Done**

• Page 9 – line 31: Mention that those numbers are valid for April 2018.

Done

**Reply (in blue) to Referee #3**

We thank the referee #3 for the positive comments and for the detailed review of the paper.

Our reply is included after the referee comments.

Mateer et al. (1971) first proposed a Lambertian Equivalent Reflectivity (LER) concept in BUV total ozone retrievals to account for combined spectral dependence of surface, aerosols and cloud reflectance [Mateer, et al., 1971]. The concept works well because ~90% ozone is in the stratosphere, above effective reflecting surface. The simple LER concept with some modifications (e.g. extrapolated LER spectral dependence, effective surface pressure) has been successfully used in heritage (TOMS, GOME, SCIAMACHY) and current (GOME-2, OMI, OMPS, S5P/TROPOMI) stratospheric ozone and other trace gases (e.g., volcanic SO2) BUV retrievals. The need for satellite retrievals of tropospheric ozone and other pollution gases (NO2, SO2, HCHO) in partly cloudy scenes, with peak concentrations in or just above the planetary boundary layer, required modification of the simple LER concept, replacing it with the mixed-LER (MLER) concept: mixing two LER surfaces, one at the ground and the other at the effective cloud pressure, e.g., [Ahmad, et al., 2004; Stammes, et al., 2008]. The MLER approach is currently used in operational BUV pollution gas retrievals (e.g., [Levelt et al., 2018] and references therein). The MLER approach requires a-priori "clear-sky" LER estimate, which can be taken either from concurrent satellite measurements (e.g., OMI geometry-dependent GLER product uses higher-resolution atmospherically corrected MODIS BRDF [Vasilkov et al., AMT 2017]) or from prior measurements (e.g., OMI cloud-cleared climatological LER [Kleipool et al., JGR 2008]). The climatological "clear-sky" LER estimation is less accurate, since it disregards the observational geometry- and timedependence of surface reflectance. The paper by Loyola et al. presents new geometrydependent (GE-LER) LER implementation, the "Full Physics - inverse Learning Machine (FP\_ILM)" algorithm and the multiple day gridded LER product (G3\_LER) derived from the present and previous clear-sky scenes observed by S5P/TROPOMI. In previous LER implementations for ozone retrievals, the LER values were derived at non-absorbing wavelengths (e.g., 340nm and 380nm for Nimbus-7 TOMS) and spectrally interpolated to the ozone and SO2 retrieval windows. The important advantage of the new GE-LER retrieval is that it is retrieved in the same spectral fitting window used by ozone retrieval (325-335nm), thus does not require spectral extrapolation. This is the first simultaneous retrieval of both ozone and LER in this spectral window. The G3 LER can be applied to existing S5P aerosol, clouds and trace gas algorithms by replacing climatological clear-sky LER with the new G3\_LER product. I recommend publishing the paper with clarifications and technical corrections and releasing the new S5P GE\_LER and gridded G3\_LER products for community evaluation.

We include now references to Mateer et al. and Ahmad et al. in the Introduction.

**General comments**

1) The name "full physics" is misleading, because the forward radiative transfer model used for NN training does not include important physical processes, such as , aerosols and inelastic (RRS) scattering;

The goal is to retrieve the surface properties under clear-sky conditions, therefore the RTM simulations don't consider modelling of aerosols or clouds.

The impact of using RSS in the forward simulation for the GE\_LER retrieval in the ozone fitting window is negligible. We add the following in Section 4.1 *"The mean difference in GE\_LER retrievals based on LIDORT-RSS and VLIDORT is in the range of 5e-5 for SZA*<75° *and 3.5e-4 for larger SZA"*.

2) acknowledge that BRDF effects on trace gas retrievals cannot be modeled exactly using forward RTM with Lambertian surface. Estimate the ozone errors due to Lambertian surface assumption (GE\_LER or simple LER) using BRDF supplement available in VLIDORT RTM.

The following sentence is included at the end of Section 4.1 "*The BRDF effects on the ozone fitting window are well modelled using the GE\_LER approximation, the difference in the total ozone retrieved using VLIDORT and VLIDORT-BRDF simulations is in the order of 0.5 DU or 0.2%*".

3) Provide more details about GE\_LER algorithm:

a. Do you assume that GE\_LER is wavelength independent within DOAS fitting window?

Correct, we listed this assumption in Section 2.3.

b. Give reference to the machine learning (NN) software and explain selecting optimal NN topology used in the algorithm training.

We use the MATLAB neural network Toolbox. The following explanation is included in Section 4.1 "*Different NN topologies were tested using one, two, and three hidden layers*".

c. Clarify whether the RTM with Lambertian surface or with BRDF model was used for training?

As already indicated in Section 4.1, we use the VLIDORT model with Lambertian surface.

d. Explain which cloud masking algorithm was used in creating G3\_LER clear-sky daily map

We add the following explanation in Section 4.3 "we use the S5P OCRA and the VIIRS/SNPP (flying in constellation with S5P) cloud fractions fc for identifying clear-sky measurements."

e. Fig. 1– clarify that "simulated features" are DOAS ozone slant columns and polynomial closure coefficients.

Fig. 1 is the general scheme for the FP\_ILM training phase. The particularities for each GE\_LER step (e.g. VLIDORT used as forward model, NN used as machine learning, DOAS used as feature extraction) are described in Sections 2.1 to 2.4.

f. Fig. 2 – clarify that "extracted features" are DOAS ozone slant columns and polynomial closure coefficients.

Fig. 2 is the general scheme for the FP\_ILM retrieval phase. The "extracted features" used in each case are algorithm dependent, for example for the GE\_LER retrieval we use the DOAS results and for the SO2 layer height retrieval we use principal components.

4) Clarify what are effects of UV-absorbing aerosols (dust or smoke) on GE\_LER?

Absorbing aerosols can induce GE\_LER values lower than the actual surface LER. As already mentioned in Section 4.3, in the future we plan to use the S5P absorbing aerosol index for filtering the affected measurements.

5) Clarify that the neural network is trained on synthetic clear-sky spectra, but applied to the TROPKMI measurements over mixed, partly cloudy scenes (equation 5).

The GE\_LER retrieval is applied to all TROPOMI measurements. Equation 5 indicates only how we compute the effective surface height in case of cloud contamination.

6) Compare TROPOMI GE\_LER retrievals with the traditional LER retrievals at 340nm, where ozone absorption is negligible. Add TROPOMI simple LER340 map to Figure 10.

We include a new Section 4.4 describing the comparison with GOME-2 and OMI LER.

7) Publicly release G3\_LER data set for community evaluation.

The retrieved GE\_LER and the G3\_LER used for each single TROPOMI ground pixel will be included in the operational S5P total ozone product. All operational S5P products are open and free available. We will discuss with ESA/EU the possibility of disseminating the G3\_LER total ozone daily maps in the same way as the operational S5P products.

**Technical comments**

Table 2 is not mentioned in the text. reference added in Section 4.2

P1, 12: with a significant[ly] lower spatial resolution . . . corrected

13: satellite viewing [geometry] dependencies added

P2,

1: are mayor [major] error sources – clarify that the surface reflectance has larger influence on boundary layer trace gases retrievals and much less on the mid-and upper-tropospheric constituent retrievals.

corrected and clarification added.

13: significant[ly] lower spatial resolution corrected

18: (b) the effect of surface reflectance anisotropy [is] corrected

20: Retrieval of [Lambertian] effective scene albedo has been used in total ozone algorithms from nadir and limb – add pioneering reference: Mateer et al., 1971. corrected. Reference to Mateer et al. added two sentences before.

22: - add references to heritage TOMS ozone, e.g., Bhartia et al., 1996 McPeters, et al., 1998. - and OMI ozone references, e.g., McPeters, et al., 2015 or Veefkind, et al., 2006. added references to Bhartia (TOMS) and McPeters (OMI)

24: from other [higher spatial resolution] satellite sensors added

28:" needed for computing LER from [and] BRDF may not be fully compatible" – need clarification: In Vasilkov et al., [2017] LER is calculated from the RT model simulated TOA radiance in a standard way, which is fully compatible with OMI cloud and NO2 retrievals. However, MODIS BRDF product may use different RT assumptions. modified to *"needed for computing MODIS BRDF may not be fully compatible"*

**P3,**

16: errors could be large and [multi-dimensional interpolations are] time consuming. . modified to "the interpolation/extrapolation in this multi-dimensional space are computational expensive, and the interpolation/extrapolation errors could be significant"

21: During the last years we [Recently] we developed an approached called . . . modified

22: applied for retrieving [ozone] profile shapes . . . added

P4,

4, . . . the surface properties - clarify what properties? Did you use RTM with Lambertian surface for training or did you use RTM with BRDF model? Specify, which land/ocean BRDF model/dataset was used for training ?

clarification added "Lambertian surface properties"

15 resolution to resolve [absorbing] features added

16 usually contains [hyperspectral] radiances at a high-dimensional space added

17 . . . avoiding the effects of the curse of dimensionality ? – clarify sentence deleted

27 Explain where does the GE\_LER information come from (i.e., equation (3))? at the end of Section 2.1 (same page as equation (3)) it is already indicated that surface properties Ae are the source of the GE\_LER

P5,

19... effective scene approximation - add reference ([Mateer et al., 1971, Coldewey-Egbers et al., 2005]) added

21 whereas a [clear-sky] LER is needed added

**22 GE\_LER retrieved under clear sky conditions – explain cloud masking algorithm explanation included**

24, Fig 3... based on the [GE\_]LER data from previous days – Clarify if the GE-LER map instrument and viewing geometry specific?

The sentence after this already explains that the G3\_LER map should include the viewing geometry dependencies. The GE\_LER is instrument specific as it is based on L1 measurements of a given instrument.

25-26 (BRDF) effects, as it is based on radiative transfer model simulations using the actual viewing geometry – clarify did you use RTM with Lambertian or BRDF surface? What surface BRDF model/dataset (if any) was used in creating training spectral dataset? RTM with Lambertian surface is used, see also reply to comment P4/4.

**P6,**

2 fitting a polynomial of clear-sky LERs averaged as function of  $\delta$  IIJC . – Please, clarify: - sentence reformulated as follows: "the dependency on the solar zenith angle can be characterized by fitting a polynomial (or exponential) function over clear-sky LERs sorted as function of  $\theta$ "

should BRDF function also depend on solar and azimuthal angles in addition to satellite view angle?

- Provide examples (add figure) of the clear sky LER(theta) for land and water surfaces. this explanation is added "solar zenith angel dependencies can be ignored when combining GE\_LER data from Sun-synchronous satellites over the same position because the angle of sunlight upon the Earth's surface is consistently maintained. Likewise relative azimuth angle dependencies are negligible in the UV"

17 synthetic UV spectra – clarify that spectra were simulated assuming Lambertian surface, no aerosols and no inelastic RRS effects. see reply to General comment 1)

19 ozone [profile?] climatology added

24 Figure 4 shows the optical densities difference – clarify definition of the optical density and the OD difference. Explain why is Figure 4 necessary?
25 ... albedo of 0.05, 0.3, 0.6, and 0.9 [,which] correspond to water,.. – not clear how [ozone?]

**optical density is related to the surface albedo?**

clarification added "optical densities of the DOAS polynomial in Equation (2)" Fig. 4 nicely illustrate how the optical densities of the DOAS polynomial change for different conditions

28 higher [longer?] wavelength. corrected

P7,

1, Fig 4... optical density increases when the viewing zenith angle decreases – please, explain. The ozone optical density is proportional to the slant column ozone amount, which should decrease when the viewing zenith angle decreases... for all cases, the optical density increases along the wavelength region –Explain why is this important? clarification added *"optical densities of the DOAS polynomial*

 $3\ldots$  is reorganized according to (3) – clarify the meaning of equation (3) an reorganization algorithm

sentence reformulated to "The simulation results from (3) are reorganized by grouping as input the DOAS polynomial coefficients and ozone slant column, the viewing geometry, and surface height"

5... using a NN with a topology of 9-20-8-2-1, - provide reference to the NN software used and how the optimal topology has been selected? see reply to General comment 3b

10, Fig.5...represents the inverse function [of the synthetic dataset] in a very precise way – this does not guarantee similar accuracy when applied to the real satellite measurements. we agree

Figure 6(a) title and color bar show "E\_LER" – change to GE\_LER Figure 6(b) – explain cloud fraction stripes over Antarctica? The cloud stripes over Antarctica are an artefact of the S5P v1 cloud retrieval algorithm that is based on OMI cloud-free composites and scan angle corrections. The S5P v2 of the cloud algorithm solves this issue.

20 In the case of clear-sky ( $\delta$ 'IS ,S $\delta$ ''IS' R'  $\leq$  0.05 ) the GE\_LER represents the surface albedo – clarify if GE\_LER represents hemispherical albedo or directional BRF ? clarification added *"hemispherical surface albedo"*

25 the TROPOMI clear-sky GE\_LER and OMI LER climatology - Add comparison with the

OMI/TROPOMI simple LER at 340nm in Table 2. 26 summarized in Figure 7. - in Table 2? We include a new Section 4.4 describing the comparison with GOME-2 and OMI LER.

P8,

1... aggregating normalized [GE\_]LER from the couple of days. – these retrievals are obtained under different viewing geometries. - Couple of days may not be sufficient to obtain cloud-free observations over certain locations. - Explain how GE\_LER are normalized and what viewing geometry does the aggregated G3\_LER map correspond to? sentence reformulated. explanation added "normalized to the central detector pixel (nadir)"

10... averaged as function of the viewing zenith angle. – BRDF depends also on solar zenith and relative solar azimuthal angles. Why is this dependence ignored? see reply to P6, 2

Fig. 8 Why is sun-glint is not visible for the water surface GE\_LER and "hot spot" is not visible for the land GE\_LER ? What would GE\_LER look like for a cloud-free sun-glint region? as already explained in the second sentence of 4.3, measurements affected by sun-glint are not used in the G3\_LER

Fig. 9(a) – what viewing geometry does the aggregated G3\_LER map corresponds to? Reduce upper scale or use logarithmic scale to better show LER variability for snow-free regions. nadir, see also reply to P8, 1

Clarify wavelength for the OMI climatological LER. "(335 nm)" added

Fig 9 caption: the ma[j]or differences corrected

Fig 10. Add comparison with the TROPOMI simple LER map at 340nm (negligible ozone absorption) TROPOMI LER at 340 nm is not available

25 associated to [with] the coarse resolution corrected

26 most important[ly] corrected

p9,
5 what is even wors[e]
corrected

11 reduced from -2.53  $\pm$  2.46% using OMI LER to 0.78  $\pm$  3.49% using TROPOMI G3\_LER - why did the standard deviation increase? it was a typo, the correct value should be 2.49

P12,

11 Loyola, D., et al.: The near-real-time total ozone retrieval algorithm from TROPOMI onboard Sentinel-5 Precursor, Atmos.Meas. Tech. Discuss., in preparation, 2019. –provide complete citation done

**Applying FP\_ILM to the retrieval of geometry-dependent effective Lambertian equivalent reflectivity (GE\_LER) daily maps fromto account for BRDF effects on UVN satellite measurements of trace gases, clouds and aerosols**

**5 Diego G. Loyola1, Jian Xu1, Klaus-Peter Heue1, Walter Zimmer1**

1German Aerospace Centre (DLR), Remote Sensing Technology Institute, Oberpfaffenhofen, 82234 Wessling, Germany

Correspondence to: Diego Loyola (Diego.Loyola@dlr.de)

Abstract. The retrieval of trace gas, cloud and aerosol measurements from ultraviolet, visible and near-infrared (UVN) sensors requires precise information on the surface properties that are traditionally obtained from Lambertian equivalent reflectivity (LER) climatologies. The main drawbacks of using such-LER climatologies for new satellite missions are (a) climatologies are typically based on previous missions with a-significantly lower spatial resolutions, (b) they usually do not fully take into account the fully for satellite viewing geometry dependencies characterized by the bidirectional reflectance distribution function (BRDF) effects, and (c) climatologies may differ considerably from the actual surface conditions

15 especially under\_with snow/ice situationsscenarios.

In this paper we present a novel algorithm for the retrieval of geometry-dependent effective Lambertian equivalent reflectivity (GE\_LER) from UVN sensors; the algorithm is based on the full-physics inverse learning machine (FP\_ILM) retrieval. The rR adiances are simulated using a radiative transfer model that takes into account the satellite viewing geometry and the inverse problem is solved using machine learning techniques to obtain the GE\_LER from satellite measurements.

- The GE\_LER retrieval is optimized not only for the trace gas retrievals employing\_using the DOAS algorithm, but also for and the large amount of data from existing and future of the new atmospheric Sentinel satellite missions. The GE\_LER can either be deployed\_used directly for the computation of AMFs using the effective scene approximation or it can be used to create a global gapless geometry-dependent LER (G3\_LER) daily map <del>can be easily created</del> from the GE\_LER under clearsky conditions for the computation of AMFs using the independent pixel approximation.
- The FP\_ILM-GE\_LER algorithm is applied to measurements of TROPOMI launched in October 2017 on board the EU/ESA Sentinel-5 Precursor (S5P) mission. The TROPOMI GE\_LER/G3\_LER results are compared with climatological OMI and GOME-2 LER datasets and the advantages of using GE\_LER/G3\_LER are demonstrated for the retrieval of total ozone from TROPOMI.

**6. Introduction**

Lack of knowledge of the magnitude of Uncertainties about the surface reflectance and the neglect of surface not accounting their anisotropic effects properties are the two majyor error sources for the retrieval of trace gas, cloud and aerosol information from ultraviolet, visible and near-infrared (UVN) satellites measurements (Vasilkov et al., 2018; Lorente et al., 2018; Lin et al., 2014; Seidel et al., 2012; Zhou et al., 2010). Surface reflectance has a stronger influence on the retrievals of

- 5
- boundary layer trace gases and aerosols than is the case for mid- and upper-tropospheric trace gas and cloud retrievals. For example errors of 0.02 in the surface reflectivity may induce errors of 10%–20% in retrieved SO2 column amount (Lee et al., 2009) and seasonal snow cover <del>could</del>-can change the retrieved NO2 column by 20%-50% (O'Byrne et al., 2010) and the retrieved O3 column by 5%–35% (Lerot et al., 2014).

The Lambertian Equivalent Reflectivity (LER) concept was first introduced for the BUV (Backscatter Ultra-Violet) total

- 10 ozone retrievals (Mateer et al., 1971) and it was extended to retrievals of tropospheric ozone, NO2, SO2 and other pollutants under partly cloudy conditions using the independent pixel approximation (Ahmad et al., 2004). Traditionally, surface properties are obtained from Lambertian equivalent reflectivity (LER) climatologies and in the case of new missions like such as TROPOMI launched in October 2017 on board the EU/ESA Sentinel-5 Precursor (S5P) mission, the climatologies used at the beginning of the mission start are based on LER data from previous missions like such as TOMS (Herman and
- 15 Celarier, 1997), GOME (Koelemeijer et al., 2003), OMI (Kleipool et al., 2008), SCIAMACHY (Tilstra et al., 2017), and GOME-2 (Pflug et al., 2008).

The unprecedented spatial resolution of TROPOMI (3.5 x 5.5 km2 currently and 3.5x7 km2 for data before August 6th 2019) has clearly showed shown the disadvantages of using LER climatologies based on previous missions with a significantly lower spatial resolution. Indeed, The initial studies version of the TROPOMI trace gas retrieved products based on such LER

- 20 using climatologies have exhibited show flawed patterns related to the coarser resolution of the OMI LER climatology. A LER climatology based on TROPOMI measurements could solve this particular problem, but creating such a new TROPOMI LER climatology will probably require at least two years of data. Furthermore, there are two fundamental common problems with typical LER climatologies: (a) the actual surface conditions of a satellite measurement may differ considerably from climatological values-like, as for example under-for snow/ice scenariossituations, and (b) the effect of 25 surface reflectance anisotropy are-is usually not properly covered by the climatology.
  - Retrieval of Lambertian effective scene albedo has been used in total ozone algorithms from nadir and limb satellite sensors, see for example Bhartia et al., 1996 and McPeters, et al., 2015. The WFDOAS (Coldewey-Egbers et al., 2005) algorithm approach-retrieves the effective LER at 377 nm, while the GODFIT (Lerot et al., 2010) and SAGE III (Raul and Taha, 2007) approaches both retrieve simultaneously the effective LER and other parameters along with total ozone the effective LER
- 30 and other parameters.

Another approach used for NO2 and cloud retrievals involved is-the computation of LER from bidirectional reflectance distribution function (BRDF) data obtained from other satellite sensors with higher spatial resolution. In a recent work

(Vasilkov et al., 2017), the BRDF data from MODIS is first resampled to the lower resolution of the OMI instrument, and then a geometry-dependent LER is computed using radiative transfer model simulations. Unfortunately MODIS BRDF data is available only from visible (VIS) wavelengths and rescaling the VIS BRDF (or LER) to UV is not straightforward. Furthermore, the radiative transfer (RT) model assumptions needed for computing LER from MODIS BRDFs may not be fully compatible with the RT model assumptions required for UV-based made in the trace gas retrievals.

5

In this paper we present a novel algorithm to be used not only for the retrieval of geometry-dependent effective Lambertian equivalent reflectivity (GE LER) from UVN measurements but also for and-the creation of global gapless geometrydependent LER (G3 LER) daily map based on using GE LER data obtained for under 
[revised manuscript text omitted]

30 retrievals based on using theis novel product is TROPOMI G3 LER daily maps are clearly substantially superior to those

one-created using thebased on OMI\_LER climatology. The ozone fields are not only much more-smoother, but also the differences compared to the total ozone from CAMS in April 2018 is reduced from  $-2.53 \pm 2.46\%$  to  $0.78 \pm 3.49\%$  in the latitudinal region [80°S-60°S]. Large eErrors oin the S5P total ozone between -10% and +15% induced by snow/ice miss-representations in the OMI\_LER climatology are removed using with the FP\_-ILM GE\_LER/G3\_LER TROPOMI products.

- 5 FP\_ILM-GE\_LER can be applied to any trace gas, cloud and aerosol product retrieved in the UVN and is fully compatible with the DOAS/AMF settings used for the trace gas retrievals. GE\_LER and G3\_LER can be used as inputs for computing AMFs, either with based on the effective scene assumption approximation or the independent pixel approximation respectively. In this paper we demonstrated their effectiveness for improving the quality of TROPOMI the total ozone from TROPOMI; in the near future we will-plan to extend GE\_LER/G3\_LER to the fitting windows of for the S5P operational UVN cloud product (Lovola et al. 2018) and the UV/VIS trace gases NO2 (van Geffen et al. 2018). SO2 (Theys et al.
- 10 UVN cloud product (Loyola et al., 2018), and-the\_UV/VIS trace gases NO2 (van Geffen et al., 2018), SO2 (
[revised manuscript text omitted]
 NO2, Atmos. Meas. Tech., 3, 1185-1203, <a href="https://doi.org/10.5194/amt-3-1185-2010">https://doi.org/10.5194/amt-3-1185-2010</a>, 2010.

Table 4: Ranges of for the input parameters appropriate used for radiance simulations in the total ozone fitting window; the ozone profiles are classified as a function of the total column. Smart sampling is employed used to generate node points optimally covering all input dimensions and more than  $2 \times 10^5$  synthetic UV spectra are generated.

| Parameter              | Minimun | Maximum  |
|------------------------|---------|----------|
| Ozone Profile          | 125 DU  | 575 DU   |
| Solar Zenith Angle     | 0°      | 90°      |
| Viewing Zenith Angle   | 0°      | 70°      |
| Relative Azimuth Angle | 0°      | 180°     |
| Surface Albedo         | 0       | 1        |
| Surface Pressure       | 125 hPa | 1013 hPa |

Table 5: Summary of the comparison between TROPOMI GE\_LER clear-sky and OMI LER (first three rows) and between as well as for-TROPOMI GE\_LER cloudy and ROCINN\_CRB cloud albedo (rows 4-6). There are more than 4.5 million clear-sky and more than 1.4 million cloudy cases out of approximately the around-15 million S5P measurements from-in April 10th, 2018.

|                    | Number    | Mean    | Std. Dev. |
|--------------------|-----------|---------|-----------|
| Clear-sky Land     | 866 907   | 0.0014  | 0.0624    |
| Clear-sky Water    | 1 837 686 | -0.0144 | 0.0762    |
| Clear-sky Snow/Ice | 1 852 222 | -0.0048 | 0.2573    |
| Cloudy Land        | 254 645   | 0.0834  | 0.1865    |
| Cloudy Water       | 1 084 985 | 0.0487  | 0.1464    |
| Cloudy Snow/Ice    | 127 636   | -0.0343 | 0.5432    |

Table 6: Latitudinal differences between total ozone from CAMS and S5P using TROPOMI G3\_LER and OMI LER for the complete-month of April 2018. The values represent the total number of measurements for each latitudinal range and the mean differences ± standard deviations (in percentages). Latitude bands with less than 100000 data points/degree were skipped, due to the polar winter conditions, there are hardly any data south of 81°S. The number of measurements increases towards higher in the north because of the overlapping orbits.

| Latitude  | Number   | TROPOMI
C3 LER     | OMI LER      |
|-----------|----------|-----------------------|--------------|
| 80°S-70°S | 11297206 | $-1.341 \pm 2.364$    | -2.041±2.114 |
| 70°S-60°S | 29018428 | $-0.364 \pm 2.472$    | -2.727±2.300 |
| 60°S-50°S | 32351377 | $0.557 \pm 1.783$     | 0.808±1.815  |
| 50°S-40°S | 31580917 | $-0.345 \pm 1.189$    | 0.048±1.224  |
| 40°S-30°S | 31154717 | $-0.776 \pm 0.906$    | -0.336±0.930 |
| 30°S-20°S | 30948143 | $-0.726 \pm 0.770$    | -0.252±0.807 |
| 20°S-10°S | 30814933 | $-0.001 \pm 0.736$    | 0.537±0.745  |
| 10°S-0°S  | 30744238 | $-0.163 \pm 0.774$    | 0.517±0.720  |
| 0°N-10°N  | 30732173 | $-0.199 \pm 0.833$    | 0.607±0.738  |
| 10°N-20°N | 30779225 | $-0.581 \pm 0.798$    | 0.142±0.728  |
| 20°N-30°N | 30894360 | $-0.788 \pm 0.945$    | -0.097±0.901 |
| 30°N-40°N | 31091907 | $-0.710 \pm 1.340$    | 0.173±1.336  |
| 40°N-50°N | 31469922 | $-0.456 \pm 1.858$    | 0.584±1.880  |
| 50°N-60°N | 32250750 | $-0.474 \pm 1.721$    | 1.287±1.920  |
| 60°N-70°N | 39590441 | -0.977 ± 2.211 | 1.155±2.798  |
| 70°N-80°N | 56545121 | $-1.182 \pm 2.581$    | -0.730±2.701 |
| 80°N-90°N | 26178029 | $-1.717 \pm 2.424$    | -1.595±2.317 |

---

## Author Response (AR2)

**Reply (in blue) to Referee #2**

**We thank the referee #2 for the final comments.**
Our reply is included after the referee comments.

- Page 10, lines 29-30: "We conclude that the historical climatologies from OMI and GOME-2 do not properly represent actual snow/ice conditions observed in 2018/2019." This conclusion is very thin and there is no clear evidence supporting this, especially as it is ensured that pixels are well affected by snow/ice for all three instruments for the comparison. That would imply that the albedo of the snow/ice itself has changed between the current conditions and those from a few years ago, which is unlikely. I would guess that it is more related to the L1 calibration issues discussed by the authors in the reply to the reviewers, possibly leading to a high bias in the S5p albedo. In fact, if the S5p albedo was slightly lower at high latitudes, the agreement with CAMS in Fig. 13 would be slightly further improved (currently there is a low bias). Please mention in the paper the L1 version that you used and discuss the possible impact of the future L1 upgrade.

We fully agree that the current S5P L1 calibration issues may have an impact on the retrieved albedo and this could affect the comparison with other datasets, but the intention of this sentence is to emphasize that the retrieved GE_LER in contrast to climatologies can distinguish between snow/ice and normal conditions.

To clarify this point we added the following sentence "*The snow/ice information from VIIRS agrees better with the GE_LER albedo than with climatological LERs, see Figure 12.*" before "*We conclude ...*"

The L1 version and possible impact of future L1 upgrades is already mentioned in page 9, line 2: "*version 1 of the TROPOMI Level 1 product has small deficiencies in the UV .... It is expected that these issues will be solved for version 2 of the TROPOMI Level 1 product*"

- Page 2, line 7: I still disagree with the upper limit of 35% cited here. I don't see how the impact of the scene reflectivity uncertainty on the total ozone column, which is mostly located in the stratosphere can be that large. In fact, your Figure 12 gives a good estimate of the error by comparing the panel (f) and (d) and it seems to be around 10%. So 15% as an upper limit would be more reasonable.

Upper limit changed to 15%

- Page 12, line 28: "like Sentinel-5P" instead of "like Sentinle-5P".

Thanks for detecting this typo, it is corrected now